# Receptor transporter protein 4 (RTP4)-mediated repression of hepatitis C virus replication in mouse cells

**Michael P. Schwoerer[1], Sebastian Carver[1], Aaron E. Lin[1], Jianche Liu[1,2], Thomas R. Cafiero[1], Keith A. Berggren[1], Serene Dhawan[3], Saori Suzuki[1]¤, Brigitte Heller[1], Celeste Rodriguez[1], Aoife K. O'Connell[4], Hans P. Gertje[4], Nicholas A. Crossland[4,5,6], Alexander Ploss****[1]\***

**1** Department of Molecular Biology, Princeton University, Princeton, New Jersey, United States of America, **2** Zhejiang University-University of Edinburgh Institute, Zhejiang University, Haining, China, **3** Princeton Neuroscience Institute, Princeton University, Princeton, New Jersey, United States of America, **4** National Emerging Infectious Diseases Laboratories, Boston University, Boston, Massachusetts, United States of America, **5** Department of Pathology and Laboratory Medicine, Boston University Chobanian & Avedisian School of Medicine, Boston, Massachusetts, United States of America, **6** Department of Virology, Immunology & Microbiology, Boston University Chobanian & Avedisian School of Medicine, Boston, Massachusetts, United States of America

¤ Present address: Department of Virology, Faculty of Medical Sciences, Kyushu University, Fukuoka, Japan
\* aploss@princeton.edu

## Abstract

Hepatitis C virus (HCV) exhibits a narrow species tropism, causing robust infections only in humans and experimentally inoculated chimpanzees. While many host factors and restriction factors are known, many more likely remain unknown, which has limited the development of mouse or other small animal models for HCV. One putative restriction factor, the black flying fox orthologue of receptor transporter protein 4 (RTP4), was previously shown to potently inhibit viral genome replication of several ER-replicating RNA viruses. Since the murine but not the human ortholog is a potent inhibitor of HCV, we aimed to analyze the potential role for RTP4 in restricting HCV replication in mice. We demonstrated that mouse RTP4 (mmRTP4) functions as a dominant inhibitor of HCV infection. Via interspecies domain-mapping, we identified the zinc-finger domain (ZFD) of murine RTP4 as essential for inhibiting HCV, consistent with prior work. Introducing mmRTP4 into HCV-infected Huh7 cells profoundly reduced HCV NS5A protein production and virion release, demonstrating that mmRTP4 can also disrupt already established HCV replication complexes. This inhibition of HCV was not driven by induction of interferon-stimulated genes based on bulk RNA-seq, suggesting that mmRTP4 might directly act on HCV replication. Indeed, by *in situ* proximity ligation, we found that mmRTP4 directly associates with the HCV NS5A protein significantly more than human RTP4 during infection. However, disrupting RTP4 expression in mice expressing humanized alleles of CD81 and

which permits unrestricted use, distribution, and reproduction in any medium, provided the original author and source are credited.

**Data availability statement:** The bulk RNA-seq data produced in this study has been deposited to the NCBI Gene Expression Omnibus (GEO) under accession number GSE292032. All data supporting the conclusions of this study are reported in the paper. The source data are available on Princeton Data Commons via the following URL: https://doi.org/10.34770/xt1d-5c49.

**Funding:** This study was supported by grants from the US National Institutes of Allergy and Infectious Diseases (R01 AI138797, R01 AI153236, R01 AI146917, R01 AI168048, R01 AI107301, R01 AI181664 and U19 A171401 all to A.P., https://www.niaid.nih.gov/), Open Philanthropy (to A.P., https://www.openphilanthropy.org/) the Princeton Catalysis Initiative (to A.P. https://pci.princeton.edu/), the Princeton Center for Health and Wellbeing (to A.P., https://chw.princeton.edu/), and Princeton University (to A.P., https://www.princeton.edu/). This work utilized NIH S10 Shared Instrumentation Grants S10 OD026983 and S10 OD030269 (to N.A.C., https://orip.nih.gov/division-construction-instruments/s10-instrumentation-programs). M.P.S. was supported by a predoctoral fellowship from the High Meadows Environmental Institute (https://environment.princeton.edu/) and the National Institute of General Medical Sciences (https://www.nigms.nih.gov/) of the National Institutes of Health (https://www.nih.gov/) under grant number T32 GM007388. A.E.L is a recipient of post-doctoral fellowship (DRG-2432-21) from the Damon Runyon Cancer Research Foundation (https://www.damonrunyon.org/). The Molecular Biology Flow Cytometry Resource Facility is partially supported by the Rutgers Cancer Institute of New Jersey NCI-CCSG P30CA072720-5921 (https://www.cancer.gov/). The funders had no role in study design, data collection and analysis, decision to publish, or preparation of the manuscript.

**Competing interests:** The authors have declared that no competing interests exist.

occludin (OCLN) – the species specific cellular factors mediating HCV uptake – did not increase permissiveness irrespective of the immunocompetence of the mice. Collectively, our work provides detailed insights into the role of RTP4 in contributing to HCV's narrow host range and will inform downstream development of a more comprehensive small-animal model for this important pathogen.

## Author summary

There is a pressing need for a small-animal model that replicates the full course of HCV infection, both to test HCV vaccine candidates and to gain insights into the effects of long-term infection. The main roadblock to this goal is an incomplete knowledge of factors that inhibit the HCV life cycle in murine cells. In this work, we delve into a previously identified antiviral protein, RTP4, in the context of HCV infection in mice. We characterize functional domains in mouse and human orthologs and interrogate HCV infection in an HCV-susceptible, RTP4-deficient mouse. We conclude by proposing a mechanism for the species-specific inhibition of HCV in mouse cells.

## Introduction

Hepatitis C virus (HCV) is an enveloped (+)-sense single stranded RNA virus of the *Flaviviridae* family, genus hepacivirus [1]. HCV is the causative agent of hepatitis C, a disease of the liver that causes chronic infection in more than 58 million people worldwide [2,3]. Left untreated, chronic HCV infection leads to bridging fibrosis and long-term inflammation, leading to cirrhosis and ultimately hepatocellular carcinoma [4]. Though HCV infection can be efficiently cured with direct-acting antivirals, more individuals are (re-)infected than cured annually [5]. Moreover, HCV-induced liver damage and prolonged T-box expressed in T cells (TOX)-mediated exhaustion of HCV-specific CD8+ T cells (or "scarring") can be irreversible [6], complicating efforts to alleviate the health burden caused by HCV.

Many key questions regarding HCV infection, including vaccine development, co- and superinfections, and long-term immune responses, are inhibited by the lack of a small animal model for HCV infection. Unfortunately, HCV has a very narrow host tropism; although loosely related hepaciviruses have been reported in a variety of species [7], it is our current understanding that HCV only robustly replicates in human and chimpanzee hepatocytes [8]. While chimpanzees have played a critical role in defining the natural history of HCV and for testing drug and vaccine candidates, research in chimpanzees is currently banned in the US and most other countries [9,10]. Laboratory mice are the preferred small-animal model for HCV given the wide array of genetic tools and isogenic lines available, along with the robust infrastructure for their usage and husbandry [11].

The limitations of the HCV reproductive cycle in murine hepatocytes remain incompletely understood, implicating a combination of incompatible host factors and dominant restriction factors [8]. At the level of entry, murine CD302 (also known as DCL-1 and CLEC13A) and complement C3b/C4b receptor 1 like (CR1L) competitively bind HCV E2, and murine CD81 and OCLN do not support HCV E2 glycoprotein-mediated entry, respectively [12,13]. Several additional incompatibilities between virally encoded proteins and host factors limit efficient viral replication; for example, murine tripartite motif containing 26 (TRIM26) is unable to ubiquitinate HCV NS5B [14], and murine cyclophilin A (CypA) is unable to facilitate HCV replication to the same extent as the human orthologue [15]. Notably, assembly and release of infectious HCV virions are supported in mouse cells *in vitro* [16] and *in vivo* [17]. The species-specific nature of many of these factors has only recently been discovered, suggesting a yet-incomplete picture of mouse-HCV incompatibility.

In this work, we sought to explore whether the antiviral protein RTP4 might serve as a species-specific factor restricting HCV in mouse cells but not human cells. Previously, the host factor RTP4 has been characterized in the black flying fox *Pteropus alecto* (paRTP4) as an interferon (IFN)-inducible, broad-acting inhibitor of ER replicating viruses [18]. In the course of analyzing paRTP4 orthologues across a variety of species, it was shown that murine RTP4 (mmRTP4) exhibited a strong antiviral effect against HCV compared to a relatively weak phenotype conveyed by the human ortholog (hsRTP4) [18]. However, the specific mechanisms by which mmRTP4 inhibits HCV infection remained unexplored.

Therefore, we sought to analyze RTP4's species-specific anti-HCV function. By generating a variety of interspecies chimeric hs/mmRTP4's, we identified the murine 3CXXC zinc-finger domain (ZFD) as the determining factor of RTP4 HCV inhibition, which is consistent with prior work on paRTP4 [18]. We further concluded that the species specificity of mmRTP4 HCV inhibition does not reside in the N-terminal-most segment of the ZFD. We additionally found that expression of mmRTP4, but not hsRTP4, abrogates an established HCV infection *in vitro*, preventing the formation of infectious virions. Expression of mmRTP4 in HCV-infected human cell lines resulted in a distinct transcriptional phenotype at 12 hours post-transduction. Indeed, this response not only lacked hallmark IFN-stimulated genes (ISGs), but also contained a number of factors that have not been implicated in antiviral responses to HCV; this transcriptional phenotype resolved by 48 hours post-transduction. Given that the frequency of NS5A$^+$ cells following mmRTP4-transduction decreases well past 48 hours post-transduction, the data suggest that the extended impact of mmRTP4 upon HCV infection may be exclusively mediated by the mmRTP4 protein alone. Via an *in situ* proximity ligation assay leveraging tagged murine/human RTP4, we found that murine RTP4 associates with the HCV NS5A protein significantly more than does human RTP4.

We further aimed to assess the phenotype of RTP4-deficiency in an *in vivo* model. We did not observe a significant increase in HCV viremia between mice expressing humanized alleles of CD81 and OCLN entry factors with and without RTP4, regardless of the presence of a murine immune response. Nevertheless, these findings provide valuable insight into the molecular mechanisms of a newly appreciated factor in the intricate story of HCV's narrow host tropism. This work will further inform future development of mouse models for HCV infection and pathogenesis.

## Results

### Murine RTP4 is a dominant inhibitor of HCV infection in vitro

To assess the species-specific effects of RTP4 in HCV infection, we established an *in vitro* system to study human/mouse RTP4 chimeras (Fig 1A). Coding sequences for the murine and human orthologues of RTP4 (mmRTP4 and hsRTP4) were cloned into bicistronic lentiviral packaging plasmids expressing ZsGreen or DsRed2. Human Huh7 hepatoma cells were transduced with hsRTP4, mmRTP4, or both lentiviruses and assessed for HCV permissiveness. We quantified the frequencies of HCV NS5A$^+$ cells by flow cytometry within mm vs hsRTP4 singly or dually expressing cells 4 days post infection with HCV J6/JFH1 (Jc1) [19] (Fig 1B). We found that, whereas hsRTP4 does not have an inhibitory effect, mmRTP4 exerts a complete inhibition of HCV replication (Fig 1C). By assaying cells expressing both mmRTP4 and hsRTP4, we determined that the inhibitory effect of mmRTP4 is dominant to the mild phenotype of hsRTP4 (Fig 1C).

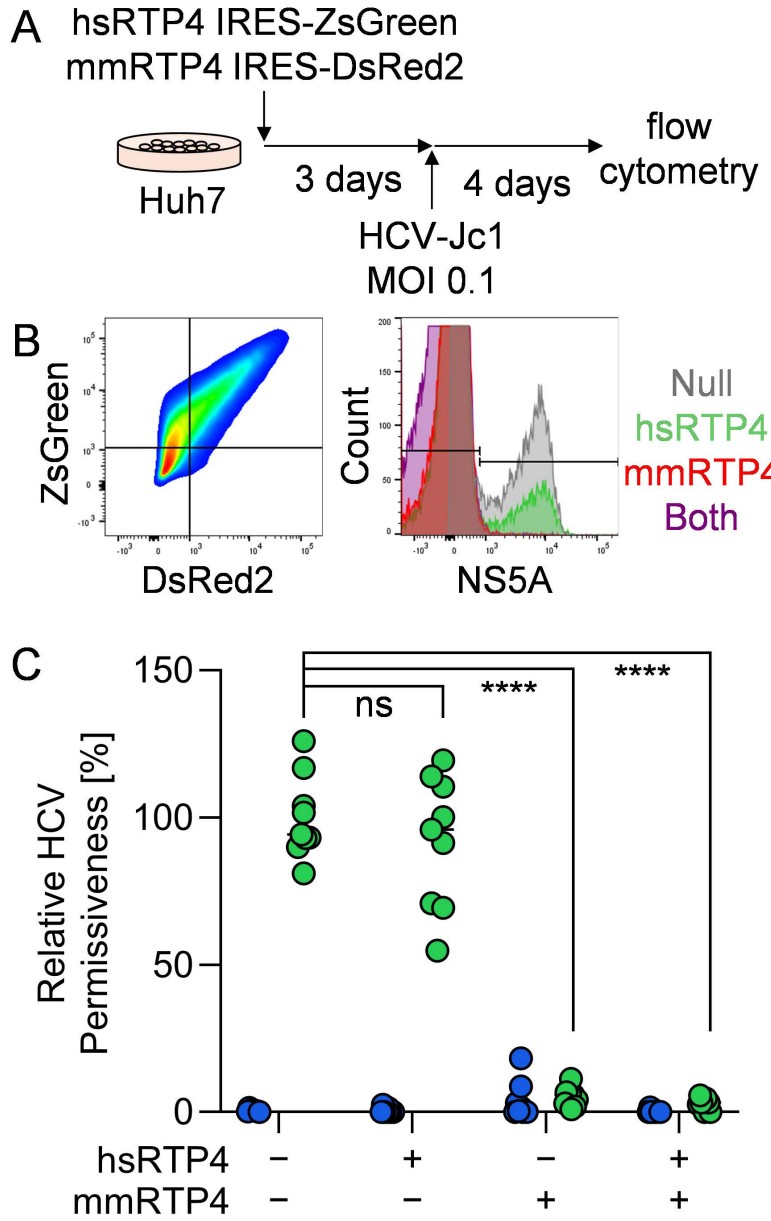

**Fig 1. Murine RTP4 exerts a dominant inhibitory effect on HCV replication. A.** Schematic of *in vitro* HCV permissiveness assay. Huh7 human hepatoma cells transduced with a lentivirus expressing hsRTP4 and ZsGreen and/or another expressing mmRTP4 and DsRed2. Three days later, cells were infected with HCV at an MOI of 0.1. Four days later, cells were trypsinized, stained with anti-NS5A, and analyzed via flow cytometry. **B.** Representative gating scheme of the flow-cytometric analysis. **C.** Quantification of the frequencies of HCV NS5A+ cells within cells expressing hsRTP4, mmRTP4 or both, relative to non-transduced cells. ****, P ≤ 0.0001; ns, not significant. Green = HCV-infected; blue = uninfected.

### The RTP4 ZFD is the determinant of species-specific HCV inhibition

Given this sharp difference in phenotypes between orthologues, we next sought to identify the regions responsible for the dominant phenotype of mmRTP4. RTP4 consists of three domains: a zinc-finger domain (ZFD), a disordered variable region (DVR), and a transmembrane domain (TM) (Fig 2A) [20]. To compare these orthologs structurally, we analyzed

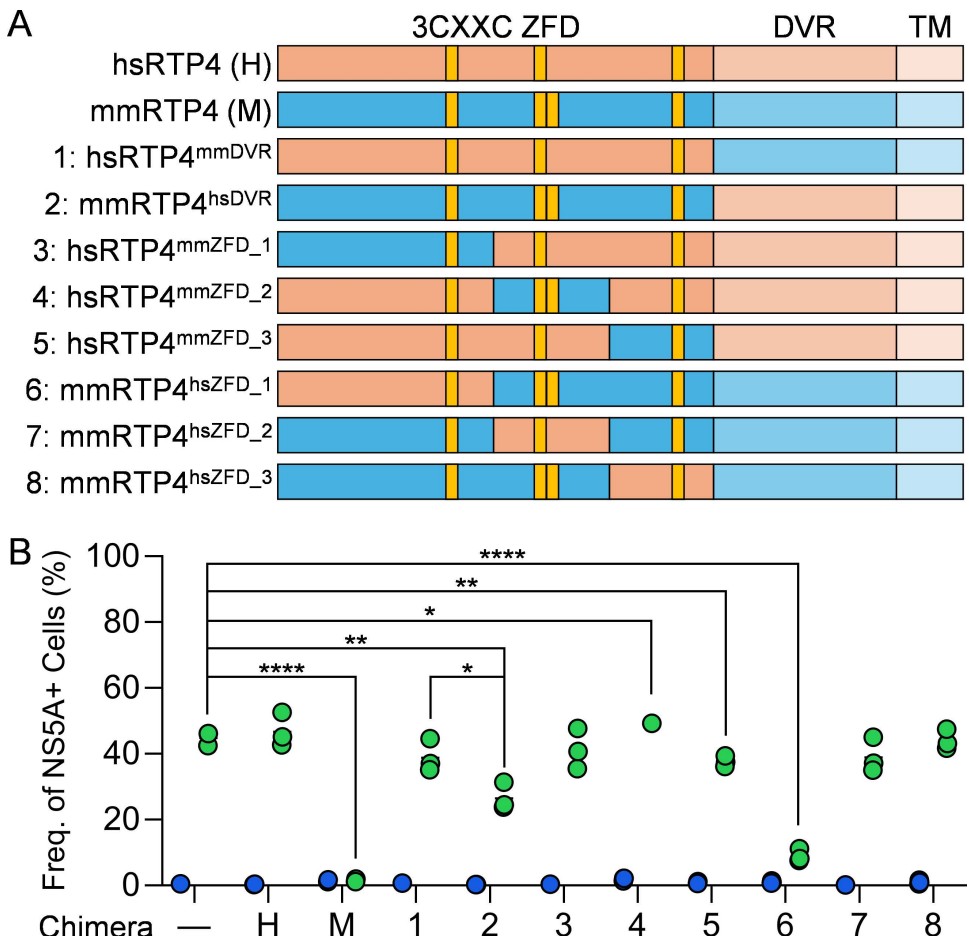

**Fig 2. The RTP4 ZFD is the determinant of species-specific HCV inhibition. A.** Schematic of chimeras generated for this study. ZFD, zinc-finger domain; DVR, disordered variable region; TM, transmembrane. Yellow boxes indicate 3CXXC motifs. **B.** Quantification of the frequencies of HCV NS5A⁺ cells within cells expressing hsRTP4, mmRTP4 or the indicated chimeras at day four post-infection. ****, P ≤ 0.0001; ***, P ≤ 0.001; **, P ≤ 0.01; *, P ≤ 0.05; ns, not significant. Green = HCV-infected; blue = uninfected.

predicted structures utilizing AlphaFold 3 [21] (S1A–S1B Fig) and observed a conservation of overall domain organization. To quantify structural similarity, we compared predicted structures utilizing the UCSF ChimeraX MatchMaker plugin [22] (S1C Fig). Both orthologs bear moderate structural similarity, with a sequence alignment score of 576.7 and root mean square deviation (RMSD) between computationally pruned structural regions being 0.763 Å (S1C Fig). Interspecies chimeras were generated by swapping murine and human DVRs (mmRTP4ʰˢᴰⱽᴿ and hsRTP4ᵐᵐᴰⱽᴿ). These chimeras showed conserved predicted tertiary structure (S1D–S1E Fig). Structural comparisons of these chimeras demonstrated conserved structural similarity (score: 586.7; pruned RMSD: 0.682 Å) (S1F Fig). *In vitro* analyses revealed that mmRTP4ʰˢᴰⱽᴿ retained an inhibitory effect, denoting the ZFD as the essence of mmRTP4 HCV restriction (Fig 2B). It must be noted that the murine DVR-TM region likely serves an auxiliary function to the murine ZFD antiviral effect, as mmRTP4ʰˢᴰⱽᴿ does not completely phenocopy the full-length mmRTP4 (Fig 2B).

The RTP4 ZFD is a 3CXXC ZFD, a subfamily of ZFDs that includes zygote arrest 1 (ZAR1) [23]. In RTP4, these include 3 CXXC motifs beginning at amino acid positions 57, 95, and 157 that facilitate binding to nucleic acids. Notably, mmRTP4 possesses an extra CXXC motif adjacent to the second CXXC starting at amino acid position 98. To probe deeper into

possible functional subdomains of the ZFD, we divided the ZFD into three regions that encompassed the separate CXXC motifs, with the exception of the overlapping pair in mmRTP4 (Figs 2A, S1G–S1L). We cloned each of these subdomains onto the backgrounds of the opposite ortholog (mmRTP4$^{hsZFD\_1-3}$, hsRTP4$^{mmZFD\_1-3}$]), and assessed their effects on HCV permissiveness (Fig 2B). hsRTP4$^{mmZFD\_1-3}$ did not impact permissiveness significantly, indicating that the complete mmRTP4 ZFD is required for inhibition. Separately, mmRTP4$^{hsZFD\_2}$ and mmRTP4$^{hsZFD\_3}$ only mildly disrupted mmRTP4's antiviral activity, whereas mmRTP4$^{hsZFD\_1}$ preserved it. This suggests either a dispensability to the N-terminus of the mmRTP4 ZFD or a sufficiency of the hsRTP4 ZFD to rescue full-length mmRTP4 activity. Nevertheless, these data provide greater insights into the species-specificity of RTP4 structural domain functions, indicating the primacy of mmRTP4 ZFD in HCV inhibition, with auxiliary functions of the DVR-TM..

## Murine RTP4 abrogates ongoing HCV infection in vitro

In previous experiments, we had expressed RTP4 in cells prior to HCV challenge, and therefore RTP4 could have reduced HCV infection by preventing viral entry and/or restricting replication. To distinguish these possibilities, we sought to assess whether RTP4 could restrict an established HCV infection. Huh7-Lunet cells [24] were infected with HCV at an MOI of 1; 7 days later, >90% of cells were NS5A$^+$. From here, we transduced HCV-infected cells with bicistronic lentiviruses expressing hsRTP4, mmRTP4, or a blank cassette (each co-expressing a bicistronic fluorophore) (Fig 3A). We quantified HCV infection by NS5A staining at a variety of timepoints thereafter, examining the frequency of infection in both RTP4-expressing and non-expressing bystander cells within each sample (Fig 3B); cells were not passaged during this time period The frequency of HCV NS5A$^+$ cells steadily decreased over the course of 9 days following mmRTP4 transduction, with no effect in hsRTP4-transduced or vehicle-transduced cells (Fig 3C).

To determine whether this decrease in infected cells correlated with infectious virus particles, we harvested and titered HCV from cell culture supernatant and cell lysates at day 9 post-transduction (cells were not passaged during this time). Titration of HCV in cell lysates revealed a significant decrease in intracellular viremia following mmRTP4 transduction compared to vehicle control (Fig 3E), whereas no such phenotype was observed in extracellular viremia (Fig 3F). These data suggest that murine RTP4 may exert a mild inhibitory effect upon infectious particle formation, albeit not to a level that would significantly decrease extracellular viremia.

## Murine RTP4 transduction induces a unique, transient transcriptional response during ongoing HCV infection

In our previous experiment, mmRTP4-expressing cells potently restricted HCV, but we were intrigued to see that bystander cells in the same dish (undetectable for the marker gene of mmRTP4) also demonstrated decreasing NS5A levels over time (Fig 3D). These data could be explained by two possibilities: either that some cells had mmRTP4 expression but were undetectable by the fluorophore marker gene, or that mmRTP4 expression might cause a non-cell autonomous effect. In other words, a cell expressing mmRTP4 directly restricts its own infection, and might simultaneously produce IFN to trigger an antiviral state in non-transduced bystander cells, restricting their infection. Indeed, some ISGs, such as RIG-I-like receptors and many IFN response factor (IRF) family proteins, both sense and amplify IFN production and sensing in a positive feedback loop, alerting neighboring bystander cells.

To test whether mmRTP4 induced an ISG response to restrict HCV, we performed bulk RNA sequencing on HCV-infected Huh7-Lunet cells transduced with mmRTP4, hsRTP4, or an empty lentivirus, collecting and analyzing samples at 12, 24, and 48 hours post-transduction, along with uninfected samples collected at 72 hours post-transduction (Fig 4A). We detected a unique transcriptomic response in mmRTP4-transduced, HCV-infected cells at 12 hours post-transduction (Fig 4B), which we did not detect in hsRTP4-transduced counterpart samples (Fig 4B). Looking more closely at the differentially upregulated genes in mmRTP4-transduced samples, we did not observe the upregulation of any ISGs or factors known to impede *Flaviviridae* infection *in vitro* (Fig 4C). As a whole, this indicates that mmRTP4 acts *per se* to inhibit HCV replication with little or no coordination of the IFN response.

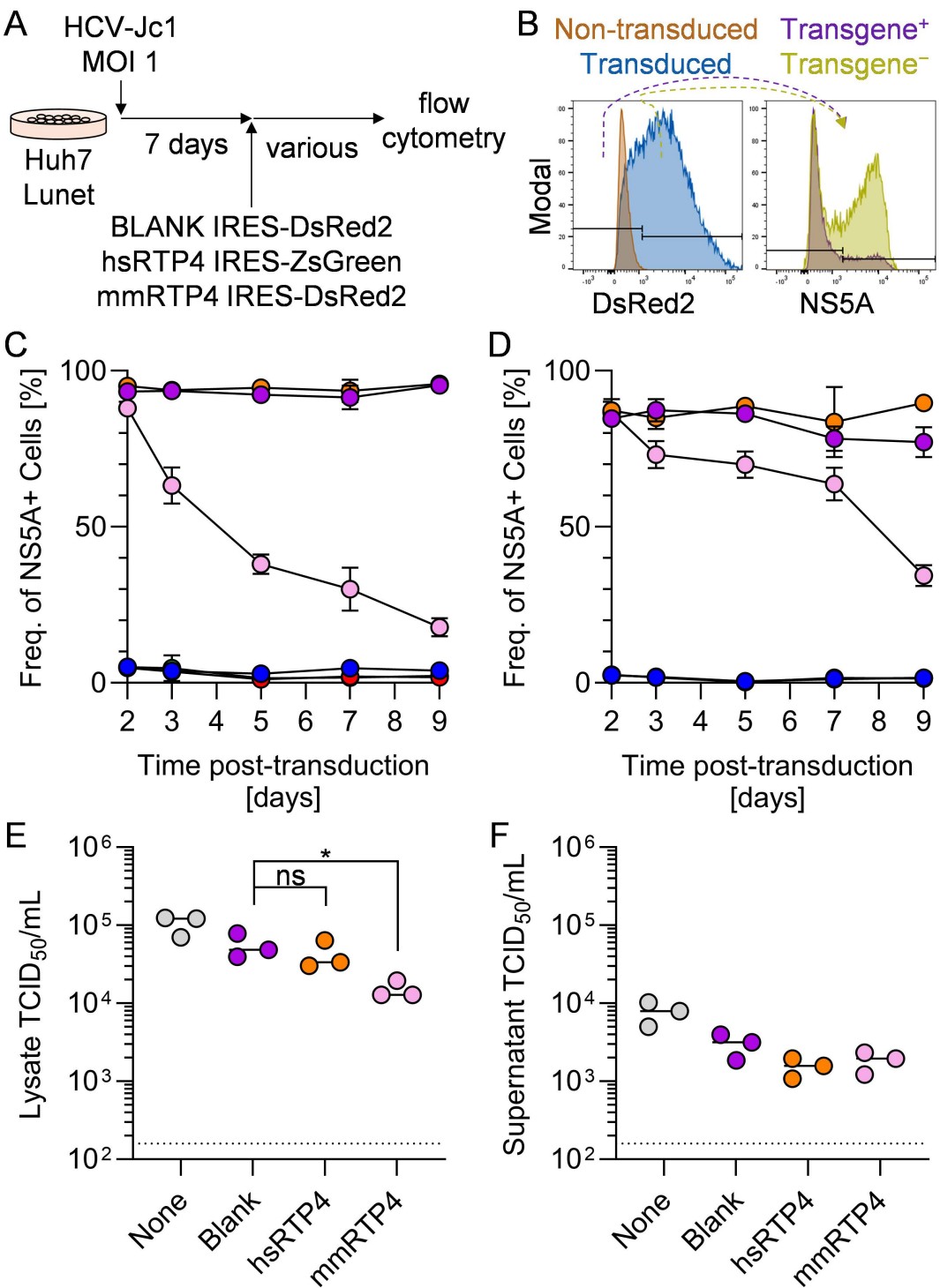

**Fig 3. Murine RTP4 abrogates ongoing HCV infection *in vitro*. A.** Schematic of experimental workflow. Huh7-Lunet cells were infected with HCV at an MOI of 1. One week later, cells were transduced with a lentivirus expressing bicistronically hsRTP4 with ZsGreen, mmRTP4 with DsRed2, or DsRed2 alone. Cells were harvested at various timepoints for analysis via flow cytometry. **B.** Representative gating scheme of the flow-cytometric analysis. **C-D.** Quantification of the frequencies of HCV NS5A+ cells within cells **C.** expressing or **D.** not expressing the given transgene. Purple = mock-transduced, HCV-infected; orange = hsRTP4-transduced, HCV-infected; pink = mmRTP4-transduced, HCV-infected; blue = mock-transduced, uninfected; red = hsRTP4-transduced, uninfected; green = mmRTP4-transduced, uninfected. **E-F.** Quantification of HCV viremia in **E.** cell lysates and **F.** supernatants at day 9 post-transduction. *, P ≤ 0.05; ns, not significant.

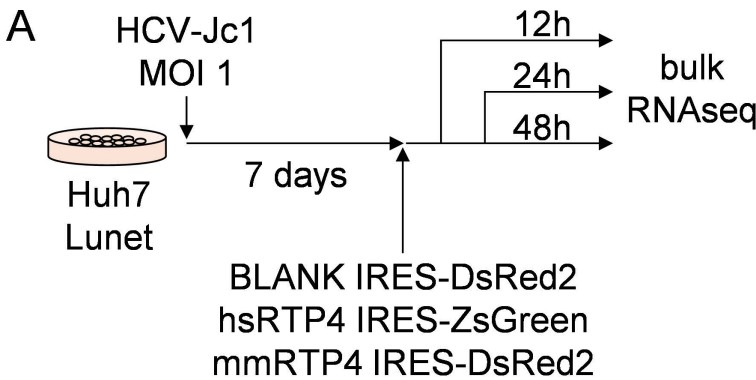

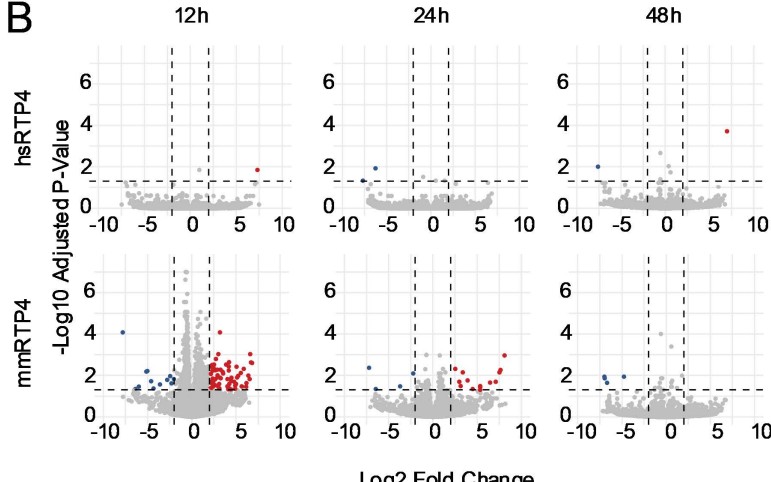

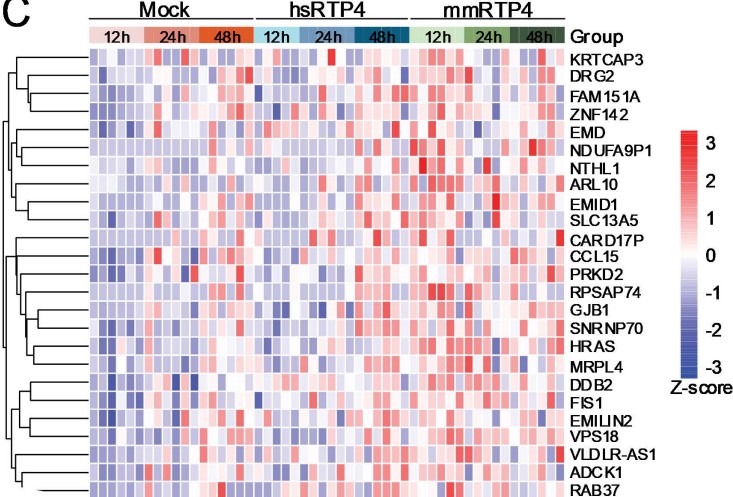

**Fig 4. Transcriptomic analysis of RTP4 transduction. A.** Sample preparation for bulk RNA sequencing. Huh7-Lunet cells were infected at an MOI of 1. One week later, cells were transduced with a lentivirus expressing bicistronically hsRTP4 with ZsGreen, mmRTP4 with DsRed2, or DsRed2 alone. Cells were harvested at 12, 24, and 48 hours post-transduction for RNA sequencing. **B.** Differentially expressed genes in hsRTP4- and mmRTP4-transduced samples, compared to mock-transduced, HCV-infected controls. **C.** Expression levels of the top 25 upregulated genes which were selected based on their adj. p-value in mmRTP4-transduced vs mock-transduced cells at 12 hours post-transduction.

## RTP4 associates with HCV replication machinery

Given that mmRTP4 did not appear to act by ISG induction, we next wondered whether RTP4 might directly inhibit HCV replication machinery. Indeed, multiple other host ISG products, such as viperin [25], VAMP-associated protein A (VAP-A) [25] and cholesterol-25-hydroxylase (CH25H) [26], are known to bind and disrupt NS5A, a key component of the viral replication complex, ultimately inhibiting HCV following innate immune activation [27]. RTP4 specifically has been demonstrated to bind a variety of protein targets depending on the host and viral contexts. In IAV infection, hsRTP4 binds directly to NS1, inhibiting its proviral interaction with RIG-I [28]. In the context of HCoV-OC43 infection, hsRTP4, but not mmRTP4, bound viral dsRNA, leading to a species-specific inhibition of HCoV-OC43 replication [29].

To determine whether such an association underpins the species-specific nature of HCV inhibition by RTP4, we generated N-terminally FLAG-tagged expression constructs for hsRTP4 and mmRTP4 (Fig 5A). We confirmed proper expression of these constructs in cell lysates (Fig 5B), as well as their localization *in vitro* (Fig 5C), and we confirmed that the tagged RTP4s phenocopied their untagged counterparts in the context of HCV infection (Fig 5D). We challenged Huh7 Lunet cells with HCV J6/JFH1-Jc1, transduced with FLAG-mmRTP4 or FLAG-hsRTP4, and then measured the ability of each RTP4 to associate with HCV NS5A. Due to the relatively low amounts of NS5A present in the infected cell, we utilized a proximity ligation assay (PLA) (Fig 5E). This sensitive, fluorescence-based approach has the benefit of providing evidence of protein-protein interactions with subcellular resolution. As expected, in the absence of HCV infection, no PLA signal was observed (Fig 5F). During infection, we established that both mmRTP4 and hsRTP4 interact with HCV NS5A protein (Fig 5F), with the murine interaction being twice as frequent (Fig 5G). These data suggest that differential association with NS5A and the HCV replication complex might be responsible for the differential inhibition of HCV by human or murine RTP4.

## RTP4 deficiency alone is insufficient to convey HCV permissiveness in vivo, irrespective of the immune status.

Finally, we aimed to determine the effect of knocking out RTP4 on HCV infection *in vivo*. To do this, we crossed a mouse line expressing minimally humanized HCV entry factors (CD81$^{EL2[H/H]}$ OCLN$^{EL2[H/H]}$), which is able to support HCV uptake but not replication, with an existing RTP4 knock-out line (RTP4$^{-/-}$) [30,31] (Fig 6A). The resultant CD81$^{EL2[H/H]}$ OCLN$^{EL2[H/H]}$ RTP4$^{-/-}$ mice, along with a control group of CD81$^{EL2[H/H]}$ OCLN$^{EL2[H/H]}$ mice, were inoculated intravenously with 1E6 tissue culture infectious doses (TCID) HCV J6/JFH1-Jc1 and monitored over the course of 20 days. HCV RNA as quantified by RT-qPCR was readily detectable in the sera of CD81$^{EL2[H/H]}$ OCLN$^{EL2[H/H]}$ (Fig 6B) and CD81$^{EL2[H/H]}$ OCLN$^{EL2[H/H]}$ RTP4$^{-/-}$ mice (Fig 6C) immediately following inoculation. However, we did not observe any increases in viral RNA at any point over the 21 day study period for either of the two genotypes. Similarly, there was not any statistically significant difference in HCV RNA levels in liver tissue harvested from these two strains at the endpoint of 21 dpi (Fig 6D and 6E) demonstrating collectively that RTP4 deficiency in HCV susceptible mice was not sufficient to increase murine permissiveness.

One hypothesis to explain undetectable viremia in these mice is adaptive immunity; it is known that depletion of CD4$^+$ cells is vital to establishing long-term RHV infection in C57BL/6 mice. To determine whether murine cellular responses effectively antagonize any putative low-level HCV RNA replication, we sought to assess the capacity of CD81$^{EL2[H/H]}$ OCLN$^{EL2[H/H]}$ RTP4$^{-/-}$ mice to sustain HCV infection in the absence of functional B, T, and NK cells. To do so, we utilized FAH$^{-/-}$ NOD Rag1$^{-/-}$ Il2rg$^{NULL}$ (FNRG) mice, a liver-injury, immunodeficient mouse platform for (xeno)transplantation of hepatocytes [32]. Hepatocytes were harvested from CD81$^{EL2[H/H]}$ OCLN$^{EL2[H/H]}$ RTP4$^{-/-}$ donor mice and injected intrasplenically into FNRG recipients (Fig 7A). Anti-FAH staining affirmed that donor CD81$^{EL2[H/H]}$ OCLN$^{EL2[H/H]}$ RTP4$^{-/-}$ (FAH$^+$) hepatocytes engrafted robustly in the parenchyma of FNRG recipients (Fig 7B). Cohorts of highly engrafted FNRG[CD81$^{EL2[H/H]}$ OCLN$^{EL2[H/H]}$ RTP4$^{-/-}$] mice were infected intravenously with 1E6 TCID HCV J6/JFH1-Jc1 and HCV RNA was quantified longitudinally in the serum over a 30 day study period.

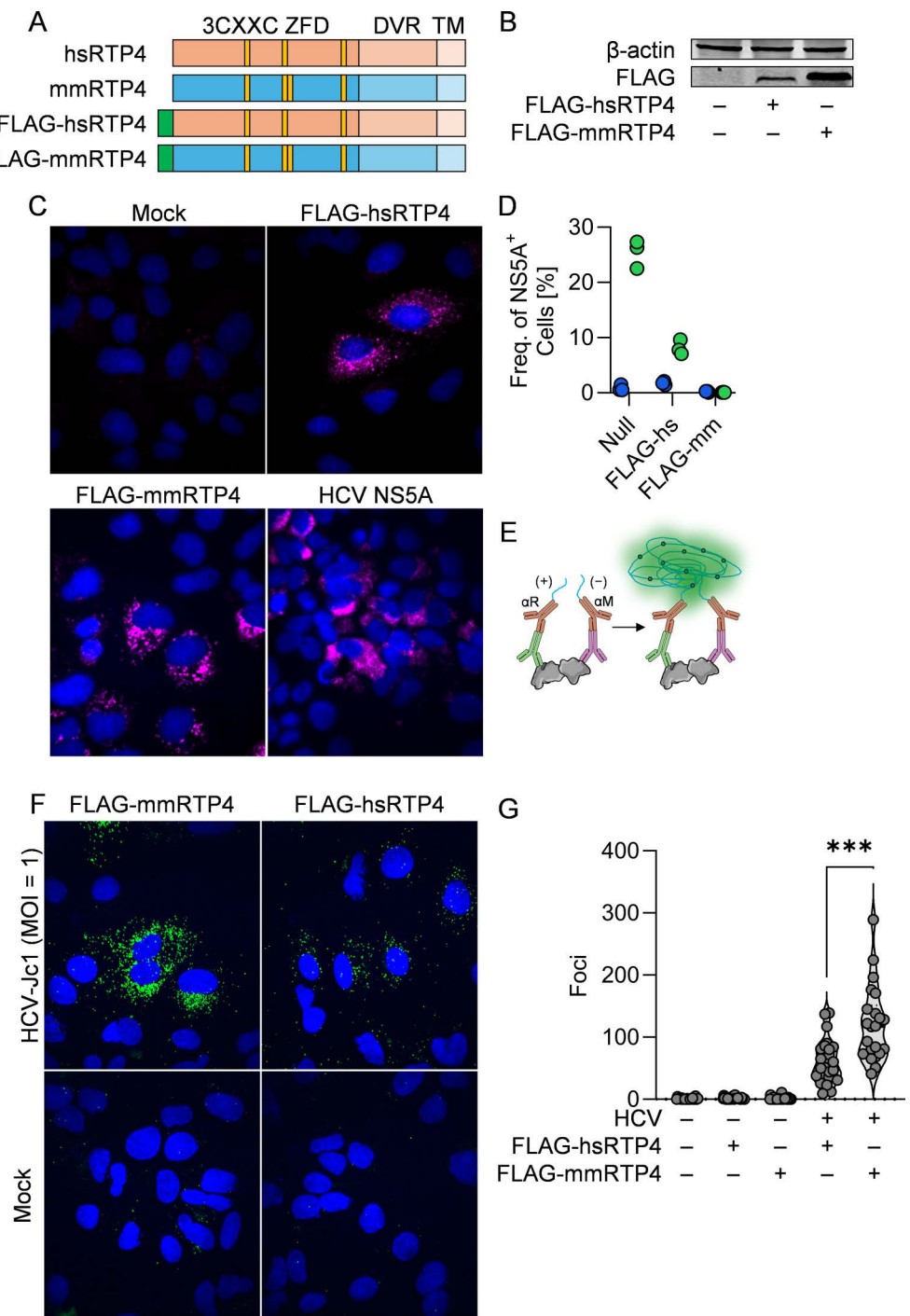

**Fig 5. Murine RTP4 associates with HCV NS5A more potently than does human RTP4. A.** Schematic of FLAG-tagged RTP4 constructs generated for this study. **B.** Western blot of FLAG-RTP4 in 293T cells transfected with overexpression constructs. **C.** Immunofluorescence staining of FLAG-RTP4 or HCV NS5A in construct-transduced or HCV-infected cells. **D.** Quantification of the frequencies of HCV NS5A+ cells within cells expressing FLAG-hsRTP4 or FLAG-mmRTP4. Green = HCV-infected; blue = uninfected. **E.** Schematic of proximity ligation assay; protein-protein interactions are leveraged for *in situ* rolling amplification with a fluorescent readout. **F.** Representative images from proximity ligation assay interrogating NS5A-FLAG(hs/mmRTP4) interactions during HCV infection or mock. **G.** Quantification of foci observed in **F**. ***, P ≤ 0.001. Some figure elements (proteins [72, 73], antibodies [74]) were sourced from the public domain and are listed as references.

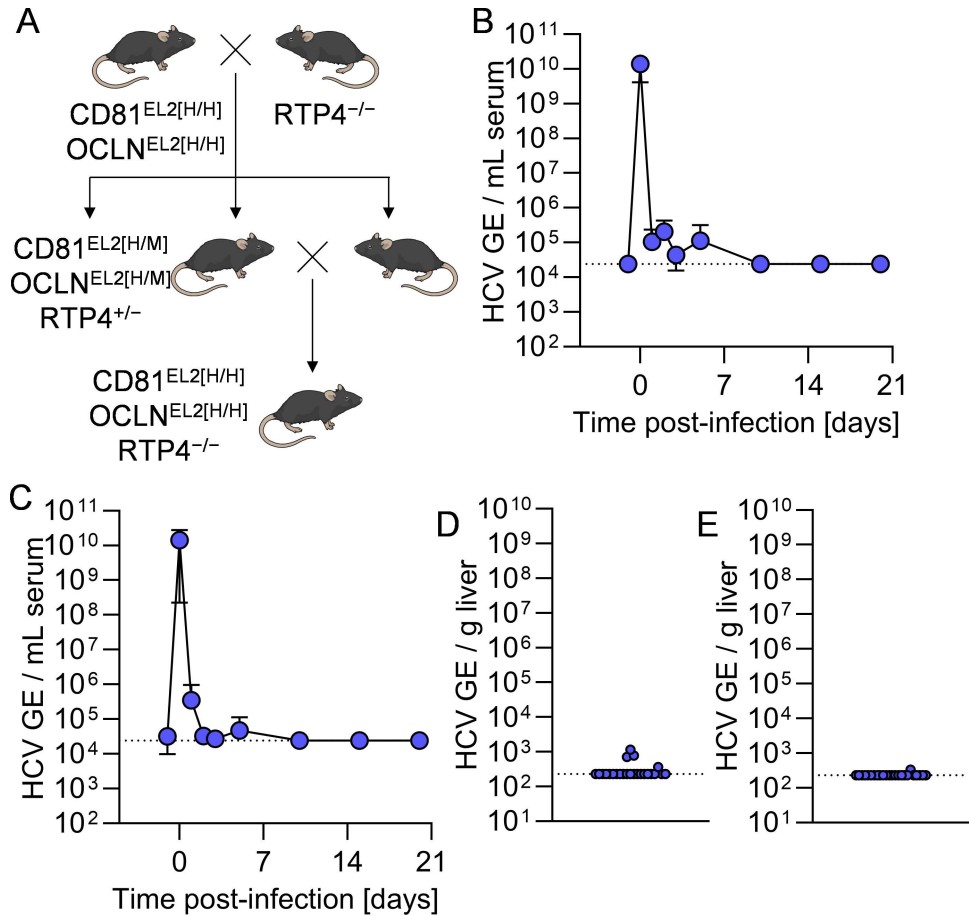

**Fig 6. HCV cannot infect HCV-susceptible mice deficient in RTP4. A.** Breeding schematic for generation of CD81$^{EL2[H/H]}$ OCLN$^{EL2[H/H]}$ RTP4$^{-/-}$ mice. **B-C.** Longitudinal HCV serum viremia in **B.** CD81$^{EL2[H/H]}$ OCLN$^{EL2[H/H]}$ and **C.** CD81$^{EL2[H/H]}$ OCLN$^{EL2[H/H]}$ RTP4$^{-/-}$ mice. **D-E.** Terminal liver viremia at day 20 in **D.** CD81$^{EL2[H/H]}$ OCLN$^{EL2[H/H]}$ and **E.** CD81$^{EL2[H/H]}$ OCLN$^{EL2[H/H]}$ RTP4$^{-/-}$ mice. GE, genomic equivalents. Some figure elements (mouse [75]) were sourced from the public domain and are listed as references.

In these highly immunodeficient mice, we did not observe any increases in HCV RNA in sera (Fig 7C) or liver tissue (Fig 7D) over the course of infection. We separately generated a cohort of mice engrafted with primary human hepatocytes, which has been demonstrated to be highly permissive to HCV infection [32,33]. We confirmed the engraftment of these mice by means of serum human albumin ELISA (Fig 7E), which correlates tightly with engraftment percentage [33]. Over the course of weeks, these mice developed sustained serum HC viremia (Fig 7F) and elevated hepatic viral load (Fig 7G). Collectively, these data demonstrate that genetic disruption of RTP4 in mice expressing human HCV entry factors is insufficient to increase their permissiveness, irrespective of the immune status of the animal.

## Discussion

Here, we probe the species-specific nature of RTP4, a gene that had recently been identified as a restriction factor of flaviviruses in the black flying fox [18]. The data from this previous work hinted that RTP4 might restrict HCV infection in mice. We found that expression of mmRTP4, but not hsRTP4, exhibits a dominant effect that can abrogate an established HCV infection *in vitro*, preventing the formation of infectious virions. By bulk RNA-seq, RTP4-mediated inhibition of HCV did not seem to trigger induction of ISGs, but instead likely acted directly on HCV replication. Via an *in situ* proximity ligation

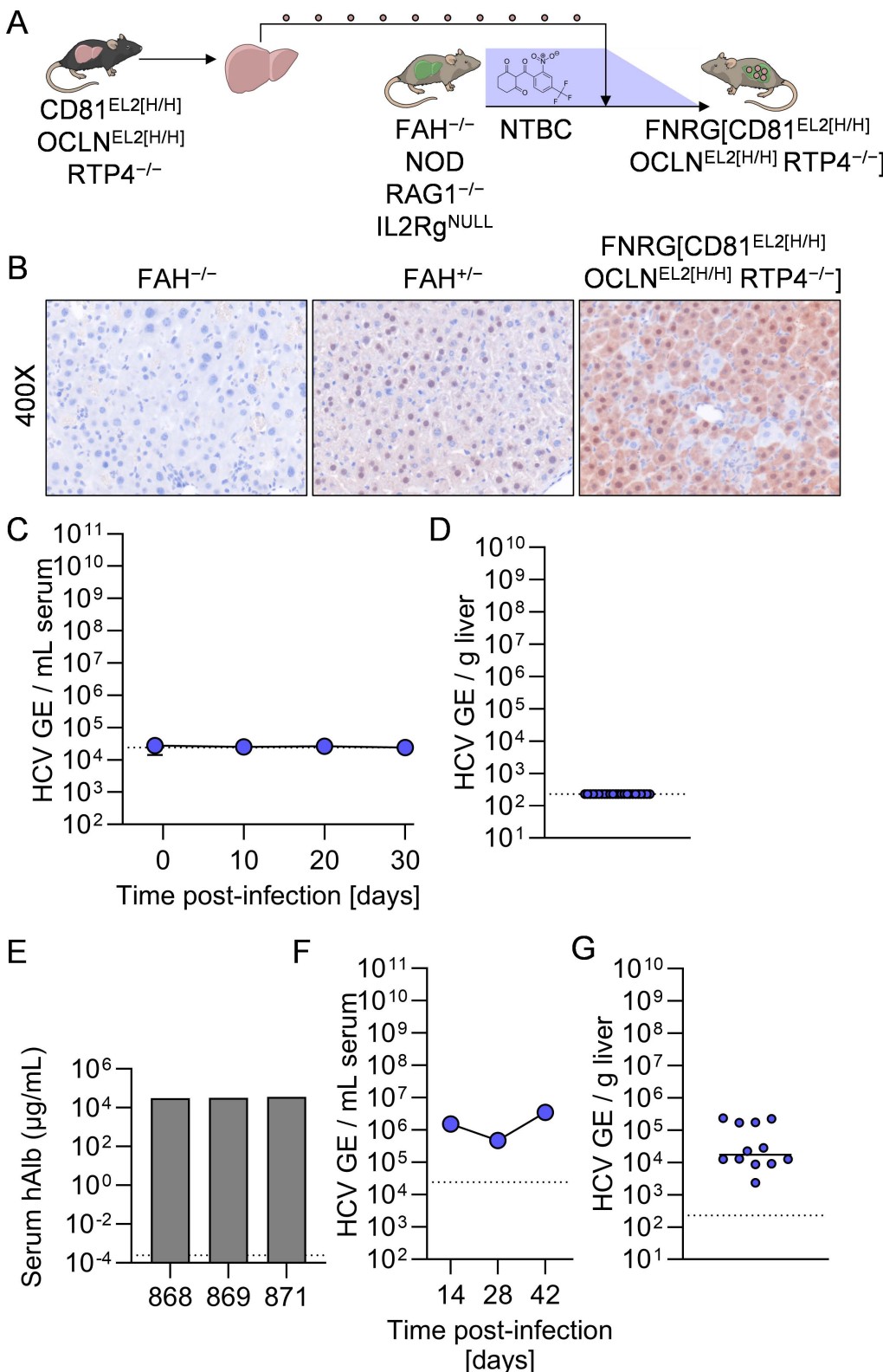

**Fig 7. HCV cannot infect immunodeficient mice harboring HCV-susceptible, RTP4-deficient murine hepatocytes. A.** Generation of FNRG[C-D81$^{EL2[H/H]}$ OCLN$^{EL2[H/H]}$ RTP4$^{-/-}$] mice. **B.** FAH (brown) staining of representative liver samples from FAH$^{-/-}$, FAH$^{+/-}$, and FNRG[CD81$^{EL2[H/H]}$ OCLN$^{EL2[H/H]}$] RTP4$^{-/-}$] mice. **C.** Longitudinal HCV serum viremia in FNRG[CD81$^{EL2[H/H]}$ OCLN$^{EL2[H/H]}$ RTP4$^{-/-}$] mice. **D.** Terminal liver viremia in FNRG[CD81$^{EL2[H/H]}$

OCLN$^{EL2[H/H]}$ RTP4$^{-/-}$] mice at day 34. GE, genomic equivalents. **E.** Serum human albumin ELISA in human liver-chimeric mice. **C.** Longitudinal HCV serum viremia in human liver-chimeric mice. **D.** Terminal liver viremia in human liver-chimeric mice at day 34. GE, genomic equivalents. Some figure elements (mouse [75], liver [76]) were sourced from the public domain and are listed as references.

assay leveraging tagged murine and human RTP4, we found that murine RTP4 associates with the HCV NS5A protein significantly more than does human RTP4. In support of this direct effect, we generated a variety of interspecies chimeric hs/mmRTP4s, and identified a C-terminal region of the murine 3CXXC ZFD as the determining factor of RTP4 HCV inhibition, which is consistent with prior work on paRTP4 [18]. Knockout of RTP4, even in immunocompromised mice, was insufficient to enable HCV replication, suggesting that RTP4 may be just one restriction factor among others that must be modified in order to convey permissiveness. In total, our study advances our knowledge on the nature, extent, and mechanism for species-specific RTP4-mediated restriction of HCV *in vitro* and *in vivo*.

Whereas prior studies leveraged a variety of truncation and point-mutated versions of paRTP4 [18] and hsRTP4 [28, 29], our study presents an analysis of interspecies domain-swap chimeric versions of RTP4 in the context of viral infection (Fig 2A). These chimeras revealed the importance of the mmRTP4 ZFD for HCV inhibition, along with the sufficiency of the hsRTP4 DVR to replace mmRTP4 DVR (Fig 2B). Notably, we found that the species-specific antiviral effect of mmRTP4 excludes the N-terminal-most section of the ZFD, which still inhibits HCV completely when humanized (Fig 2B).

What is the mechanism of action of RTP4? In its native context, RTP4 is an ISG that is strongly induced by IFN signaling. Here, we expressed RTP4 constitutively from an exogenous source (lentiviral transduction), which allowed us to isolate the effects of RTP4 separately from the rest of the ISG response. Expressing RTP4 inhibited HCV replication without triggering expression of other ISGs (Fig 4), emphasizing the extent to which this phenotype is driven solely by the expression of mmRTP4 alone. Moreover, RTP4 could restrict an already established HCV infection (Fig 3A), and mildly impact intracellular viremia (Fig 3E), both consistent with a mechanism directly targeting viral replication.

Our discovery of a direct interaction between RTP4 and the HCV NS5A protein by proximity ligation (Fig 5) further clarifies the role of this restriction factor, especially in light of other known ISGs. NS5A appears to be a common target for ISGs in mammalian cells [25,26,34]. For example, the ISG viperin directly binds both the HCV NS5A protein, an essential component for replication and immune modulation, and VAP-A, inhibiting replication [25]. Similarly, CH25H, a factor known to have multiple effector functions against HCV replication, directly binds NS5A [26,34]. These binding events likely disrupt the extremely precise structure of the replication complex, which requires a variety of HCV and host proteins for proper formation and function [35]. Just as NS5A has evolved to be a multifaceted inhibitor of antiviral signaling, so too has the host cell evolved to target NS5A with a variety of interferon-stimulated effectors. Given the established dsRNA-binding capability of human and black flying fox RTP4 in other infections [18,29], steric hindrance is a likely hypothesis for the conserved mechanism of HCV inhibition by RTP4, viperin, and CH25H. The species-specificity of mmRTP4 in inhibiting HCV replication resides in the strength of its interaction with HCV NS5A, which we measure to be two-fold higher than the hsRTP4-NS5A interaction (Fig 5G).

This species-specific restriction of HCV by RTP4 is particularly fascinating in light of the evolutionary history of this protein family. RTP4 belongs to a family of proteins that promote the cell-surface transport of G-protein-coupled receptors including bitter-taste receptors, μ-δ opioid receptors, and odorant receptors [36–38]. Variants of RTP4 have additionally been implicated in diseases ranging from primary Sjogren's syndrome to cutaneous melanoma, and it has been demonstrated to have a pro-inflammatory effect in neuroinvasive infections [30,39–41]. Evolutionarily, RTPs have undergone a significant degree of positive selection, with multiple clades evolving to serve as antiviral sentinels [42]. In mammals, hsRTP4 has evolved an antiviral spectrum against a variety of clades, showing strong effects against the flavivirus Entebbe bat virus [18], the human coronaviruses HCoV-OC43 and SARS-CoV-2 [29], and recently influenza A virus (IAV) [28]. Notably, mmRTP4 expression does not restrict infection by SARS-CoV-2 or HCoV-OC43 [29]. Conceivably, mmRTP4

underwent a similar process of evolution to target a unique range of viral clades, including hepaciviruses; mice are afflicted with a variety of murine hepaciviruses that are distantly related to HCV [7,43,44].

None of the factors described as having divergent impacts on the HCV viral cycle between human and murine orthologues are canonical ISGs [8]. This is notable, as the cell-intrinsic antiviral response has been implicated in limiting or broadening the tropism of a variety of viruses [45,46]. Our study on murine and human RTP4 thus establishes a third paradigmatic level of HCV restriction in mice. Indeed, the disruption of IFN receptors and upstream immune sentinels (MAVS, IRF1) have been leveraged to enhance the low-level replication of drug-selectable HCV subgenomic replicons and transfected full-length RNA in mouse cells *in vitro* and *in vivo* [17,47–52]. Our work here provides some explanation for this, given the dominant phenotype of murine RTP4. We note that it cannot be discounted that other IFN-stimulated factors may play a role in murine restriction of HCV similar to RTP4, given the variety of ISGs that have been described to arise during HCV infections [53].

From a practical standpoint, ablating RTP4 alone is not sufficient to serve as a mouse model for HCV. To develop such a model, it is necessary to understand the complete range of factors that prevent HCV from replicating within an immunocompetent mouse; to this end, many have been delineated, as reviewed previously [8]. In short, there are multiple factors at the levels of entry, replication, and assembly where the mouse ortholog either directly inhibits HCV's biology or is incompatible with the given viral protein. As a result, the lack of replication of HCV in CD81$^{EL2[H/H]}$ OCLN$^{EL2[H/H]}$ RTP4$^{-/-}$ and FNRG[CD81$^{EL2[H/H]}$ OCLN$^{EL2[H/H]}$ RTP4$^{-/-}$] mice (Figs 6 and 7) is not surprising. Of note, these mice, while HCV-susceptible and RTP4-deficient, express murine orthologues of several known host factors governing viral replication that are incompatible with virally encoded components of the replication complex. For example, these mice bear murine TRIM26, and are thus unable to conduct K27-linked ubiquitination of HCV NS5B [14], as well as murine CypA, which is tenfold less efficient than human CypA at facilitating viral replication [15]. Certainly, a more comprehensively humanized mouse must be generated to complete full HCV pathogenesis *in vivo*.

Recent work has broadened the scope of viral targets of human RTP4; it had erstwhile been considered only to inhibit a little-known flavivirus, Entebbe bat virus, but this has recently been extended to include human coronaviruses HCoV-OC43 and SARS-CoV-2 [29], as well as influenza A virus [28]. While murine RTP4 exerts divergent effects in the contexts of flavivirus (antiviral) and nidovirus (agnostic) infections, it is possible that it plays a role in rendering mice impermissible to other human-tropic pathogens. To this end, a version of mice expressing human RTP4 in lieu of murine RTP4 is presently being generated.

Depending on the viewpoint, the narrowness of HCV tropism is at once fortunate and frustrating. It is quite fortunate that HCV, as opposed to other members of the *Flaviviridae* family, is unable to maintain itself in a perpetual zoonotic or enzootic cycle [54]. Indeed, the restriction to humans and chimpanzees theoretically renders HCV vulnerable to full eradication should a vaccine with even partial efficacy be developed [55]. Unfortunately, this vaccine's development is hampered by, among other factors, the lack of a proper animal model for infection. Though many models have been proposed, including the use of controlled human infection models (CHIMs), a mouse model remains desirable due the ease and economy of husbandry, bounty of available tools, and broad range of ethical experiments [10,56–58]. To that end, our characterization of species-specific mmRTP4 restriction of HCV may pave the way for future immunocompetent mouse models of HCV.

## Materials & methods

### Ethics statement

All animal experiments were performed in accordance with protocols reviewed and approved by the Institution Animal Care and Use Committees (IACUCs) of Princeton University (protocol number 3063). All mice were bred and generated in the Laboratory Animal Resource (LAR) Center of Princeton University.

## Cell lines

Huh7 and Huh7.5 cells were kindly provided by Charles Rice (Rockefeller University, New York City, USA) and Huh7.5.1 cells by Frank Chisari (The Scripps Research Institute, La Jolla, USA) and Huh7-Lunet cells by Ralf Bartenschlager (University of Heidelberg, Germany). 293Tx cells were obtained from TaKaRa (Kusatsu, Shiga, Japan). All cells have been authenticated and are clear of mycoplasma contamination. All cells were maintained in Dulbecco's Modified Eagle's Medium (DMEM, Thermo Fisher Scientific, Waltham, MA, USA) containing 10% (v/v) heat-inactivated fetal bovine serum (FBS) (Bio-Techne, Minneapolis, MN, USA) and 1% (v/v) penicillin/streptomycin (Corning Inc., Corning, NY, USA) at 37 °C with 5% (v/v) CO2. Upon reaching confluence, cells were trypsinized with 0.25% (v/v) trypsin/EDTA and re-plated.

## Antibodies

The monoclonal mouse anti-NS5A 9E10 antibody [59] was generously provided by Charles Rice (Rockefeller University, New York City, USA). The following commercial primary antibodies were used: mouse anti Myc-Tag 9B11 (Cell Signaling Technology, Danvers, MA, USA), mouse anti DYKDDDDK Tag 9A3 (Cell Signaling Technology, Danvers, MA, USA), rabbit anti DYKDDDDK Tag D6W5B (Cell Signaling Technology, Danvers, MA, USA), rabbit anti HA-Tag C29F4 (Cell Signaling Technology, Danvers, MA, USA), mouse anti-β-actin (Cell Signaling Technology, Danvers, MA, USA), rabbit anti-β-actin (Cell Signaling Technology, Danvers, MA, USA). The following commercial secondary antibodies were used: goat anti-mouse Alexa 647 (Invitrogen, Waltham, MA, USA), goat anti-rabbit Alexa 647 (Invitrogen, Waltham, MA, USA), goat anti-mouse Dylight 800 (100 ng/mL, Thermo Fisher Scientific, Waltham, MA, USA), goat anti-rabbit Dylight 680 (100 ng/mL, Thermo Fisher Scientific, Waltham, MA, USA).

## Plasmid construction

Coding sequences for human and murine orthologs of RTP4 (hsRTP4 and mmRTP4) were respectively isolated from the human ORFeome library v8.1 [60] and interferon-stimulated C57BL/6 splenocytes. These were cloned into pLVX lentivirus expression constructs using restriction cloning. Chimeras mmRTP4$^{hsDVR}$ and hsRTP4$^{mmDVR}$ were generated by In-Fusion cloning. Chimeras mmRTP4$^{mmZFD[hs1-3]}$ and hsRTP4$^{hsZFD[mm1-3]}$ were synthesized as gBlocks (Integrated DNA Technologies, Coralville, IA, USA). To generate FLAG- and c-myc-tagged RTP4, forward primers were utilized to add the accordant epitope tags at the N-terminus. All constructs were subjected to Sanger sequencing to confirm proper insertions (Table 1).

## RTP4 structural prediction and analyses

For hsRTP4, mmRTP4, and all chimeric constructs generated in Fig 2, structural predictions were generated using Alpha-Fold 3 [21], and resultant structures were visualized using UCSF ChimeraX 1.6.1 [22]. Select structures were compared quantitatively using the MatchMaker plugin [61].

## Lentivirus production and transduction

Lentiviral particles encoding the given RTP4 constructs were produced by X-tremeGENE HP DNA Transfection Reagent (Roche Applied Science; Indianapolis, IN)-mediated transfection of 293TX cells seeded twelve hours prior to transfection (2E6 cells per 10 cm tissue culture dish) with 4 µg of the appropriate pLVX plasmid, 4 µg of HIV gag-pol plasmid, and 0.57 µg of the G protein of vesicular stomatitis virus (VSV-G) plasmid per transfection reaction. Supernatants were harvested at 48 and 72 hr post-transfection, stored at 4 °C and then passed through 0.45 µm membrane filters (Millipore, Darmstadt, Germany). Polybrene (final concentration of 4 mg/mL) (Sigma-Aldrich, St. Louis, MO, USA) and HEPES (final concentration of 2 mM) (Gibco, Waltham, MA, USA) were added to all lentiviral supernatants which were aliquoted and stored at −80 °C. All lentiviral transductions were performed via spinoculation with cells seeded at a concentration of 1.5E5 cells per well in a six well format 24 hr prior to transduction. Cell confluency at the time of transduction was 30–40%, and 1 mL

**Table 1. Primers utilized for generation of RTP4 overexpression constructs in this study.**

| Primer | Purpose | Sequence (5'–3') |
|---|---|---|
| PU-O-8943 | hsRTP4 FP | ATGGTTGTAGATTTCTGGACTTGG |
| PU-O-8944 | hsRTP4 RP | TCATTCTGATGTAAAGCATTTGACTACAATAAATAC |
| PU-O-8943 | hsRTP4 ZFD FP | ATGGTTGTAGATTTCTGGACTTGG |
| PU-O-10335 | hsRTP4 ZFD RP | CAAAGTGCATGCCTCACAATT |
| PU-O-10351 | hsRTP4 DVR FP (In-Fusion) | AATTGTGAGGCATGCGGCATATGTGGACAGGGCTTAAAAAGC |
| PU-O-10342 | hsRTP4 DVR RP (In-Fusion) | TTATCTAGTGAAAAGACTAAAAAGGGCAAATGCAGCAATAGACAAAAACAGAAATAGTGGTG-GCTCTATATCTGGGTCTCGACTGGGCCCTAATTTC |
| PU-O-8894 | mmRTP4 FP | ATGCTGTTCCCCGATGACTTC |
| PU-O-8895 | mmRTP4 RP | TTATCTAGTGAAAAGACTAAAAAGGGCAAATGC |
| PU-O-8894 | mmRTP4 ZFD FP | ATGCTGTTCCCCGATGACTTC |
| PU-O-10336 | mmRTP4 ZFD RP | GCATGCCTCACAATTGCGTGT |
| PU-O-10349 | mmRTP4 DVR FP (In-Fusion) | GAGGCATGCACTTTGAGTCTAAACTCTCATGGAAGATG |
| PU-O-10339 | mmRTP4 DVR RP (In-Fusion) | TCATTCTGATGTAAAGCATTTGACTACAATAAATACAAGCAGCAAAATAAAGACACAGAT-GTTCAGTGGTTCACGTTGGGGGGGGCTCTCTG |
| PU-O-10886 | FLAG-hsRTP4 FP | AAGCTTCTCGAGATGGATTACAAGGATGACGACGATAAGATGGTTGTAGATTTCTGGACTTGG |
| PU-O-10982 | FLAG-mmRTP4 FP | AAGCTTCTCGAGATGGATTACAAGGATGACGACGATAAGATGCTGTTCCCCGATGAC |

of lentivirus was added to each well. Plates were spun at 37 °C, 2 hr, 2000 rpm. Transduction efficiency was assessed via flow cytometry for all constructs (see below).

## Generation of HCV RNA and viral stocks

HCV viral RNA was produced via in vitro transcription of an XbaI-linearized J6/JFH1 (J6/JFH1-Jc1) [19] plasmid using the HiScribe T7 High Yield RNA Synthesis Kit (New England Biolabs, Ipswich, MA, USA) as outlined in the user manual. Viral RNA was purified using the MEGAclear Transcription Clean-Up Kit (Thermo Fisher Scientific, Waltham, MA, USA) following manufacturer's instructions, and quality control was performed by gel electrophoresis to ensure no significant RNA degradation. Viral RNA stocks were stored as 5 µg aliquots at −80 °C. RNA was electroporated into Huh7.5-1 cells [62]. The pellet was resuspended in the appropriate volume of ice-cold DPBS to achieve a concentration of 1.5E7 cells/mL. 6E6 cells were then electroporated in a 2 mm path length electroporation cuvette (BTX Harvard Apparatus, Holliston, MA, USA) with 5 µg of viral RNA using an ECM 830 Square Wave Electroporation System (BTX) at the following settings: five pulses, 99 µs per pulse, 1.1 s pulse intervals, 860V. Following a ten-minute incubation at room temperature, the electroporated cells were seeded into 150 mm plates and maintained in DMEM with 5% (v/v) FBS (Bio-Techne, Minneapolis, MN, USA) and 1X non-essential amino acids (NEAA) (Gibco, Waltham, MA, USA). Media was changed one day post-electroporation, and supernatants were collected twice daily from days four through six and stored at 4 °C. The pooled supernatants were passed through a 0.22 µm vacuum filter and subsequently concentrated to ~40 mL in 100 kDa MWCO Amicon Ultra-15 Centrifugal Filter Units (Millipore Sigma, Allentown, PA, USA).

## Quantification of HCV titer by limiting dilution

The TCID50/mL of concentrated virus was determined after one freeze-thaw by limiting dilution assay. Huh7.5 cells were seeded in a 96-well plate at a density of 6400 cells/well. 50 µL of ten-fold serial dilutions (from 1:1E2 to 1:1E7) of virus were added to each column of wells, with 8 wells receiving each dilution. After removal of the inoculum 6–8 hours post-infection, cells were washed with unsupplemented DMEM and cultured in 200 µL DMEM containing 10% (v/v) FBS and 1% (v/v) penicillin/streptomycin. On day 3 post-infection, cells were fixed and permeabilized in ice-cold 100% methanol

for 30 min at −20 °C. Cells were blocked in 1X PBS containing 0.1% (v/v) Tween-20, 1% (w/v) BSA, and 0.2% (w/v) skim milk for 30 min at room temperature (RT). Cells were then treated with PBS containing 3% $H_2O_2$ for 5 min at RT. Cells were then stained with a mouse anti-HCV NS5A monoclonal antibody (clone 9E10 [59], 220 ng/mL, 50 µL/well) for 1 h at RT, followed by an HRP-conjugated goat anti-mouse polyclonal antibody (Invitrogen, Waltham, MA, USA, 5 µg/mL, 50 µL/well). HRP signal was detected using DAB Peroxidase (HRP) Substrate Kit (Vector Laboratories, Newark, CA, USA). TCID50/ mL was calculated using the Reed & Muench method [63].

## Analysis of HCV infection by NS5A staining

HCV infections were conducted in a 24-well format with 3E4 cells seeded per well 12 hrs pre-infection. Infections were conducted in triplicate wells using cell-culture produced virus produced as described above. For NS5A staining, trypsinized cells were centrifuged at 2000 rpm for 3 min at 4 °C, fixed with 4% (w/v) paraformaldehyde (PFA) (Sigma Aldrich, St. Louis, MO, USA) and permeabilized in 0.1% (w/v) saponin (Thermo Fisher Scientific, Waltham, MA, USA) and 1% (v/v) FBS in DPBS. Pellets were subsequently incubated for 1 hr at room temperature with murine-produced primary antibody specific for HCV NS5A (clone 9E10) [59] diluted in FACS buffer (1% (v/v) FBS in DPBS) to 220 ng/mL. Cells were washed with FACS buffer and then incubated at 4 °C for 30 min in the dark with goat anti-mouse Alexa 647 secondary antibody (Invitrogen, Waltham, MA, USA) diluted to 8 µg/mL. Cells were subsequently centrifuged at 2000 rpm for 3 min at 4 °C, washed once with FACS buffer, and then analyzed in FACS buffer on a BD LSRII flow cytometer (BD Biosciences, Franklin Lakes, NJ, USA). All flow cytometry data were processed in FlowJo Software version 10.4.2 (FlowJo, LLC).

## Bulk RNA sequencing and analysis

HCV infections were conducted in a 10 cm plate with 1E6 Huh7-Lunet cells seeded per plate 12 hrs pre-infection. Cells were infected at an MOI of 1 and expanded over the course of a week. On day 6 post-infection, 24-well plates were seeded with 3E4 cells per well. On day 7, cells were transduced with 500 µL 1:5 diluted lentivirus as described above. At 12, 24, and 48 hours post-transduction, cells were trypsinized, centrifuged at 2000 rpm for 3 min at 4 °C, washed once with DPBS, and resuspended in 200 µL $H_2O$ with 133 µL lysis/binding buffer (Applied Biosystems, Waltham, MA, USA). Timepoints were frozen on dry-ice and stored at −20 °C until RNA extraction. RNA extraction was performed using the KingFisher Flex System (Thermo Fisher Scientific, Waltham, MA, USA). Samples were subsequently treated with 12 µL of 1X DNase I (RNase-free) (New England Biolabs, Ipswich, MA, USA) for 30 min at 37 °C. Total RNA was purified from this reaction using 0.8X SPRIselect (Beckman Coulter, Brea, CA) and quantified via NanoDrop spectrophotometer (Thermo Fisher Scientific, Waltham, MA, USA).

Up to 25 ng total RNA was used per sample for gene expression profiling. Bulk RNA-barcoding and sequencing (BRB-Seq) [64] was performed with minor modifications. At the reverse transcription (RT) step, Template Switching RT Enzyme Mix (New England Biolabs, Ipswich, MA, USA), along with a uniquely barcoded oligo(dT)30 primer was used for each sample, modified with the Illumina TruSeq Read 1 priming site instead of Nextera Read 1 [65]. The remainder of the BRB-Seq protocol was followed, pooling up to 24 first-strand cDNAs into a single tube, followed by Gubler-Hoffman nick translation cDNA synthesis, and tagmented cDNA with in-house-produced Tn5 [66]. cDNAs were amplified for 14–15 PCR cycles using a P5-containing primer and a distinct multiplexed i7 indexing primer (Chromium i7 Multiplex Kit, 10X Genomics, Pleasanton, CA). Size-selection was performed using 0.65X SPRIselect (Beckman Coulter, Brea, CA), and libraries were sequenced on one flowcell of a NovaSeq SP v1.5 flowcell (Illumina, San Diego, CA) with 28 cycles Read 1, 8 cycles Read i7, and 101 cycles Read 2.

Reads were demultiplexed with Picard v2.25.6 (from within viral-core v.2.3.1) using Q20M1 mismatch tolerance and the read structure flag '5S8B15M8B101T' in order to simultaneously process the within-pool sample barcode (from the RT primer) and the pool barcode (from the i7 indexing primer). Next, reads were mapped to the human genome hg38, plus

the HCV/Jc1 isolate reference (derived from GenBank OQ726018.1), with STAR v2.7.11b, with uniquely mapping reads counted to the comprehensive gene annotation on the primary assembly with htseq-count v2.0.5.

## Data Processing and Differential Expressed Gene (DEG) analysis

Transcript read counts for mmRTP4/hsRTP4- or mock-transduced cells at various time points post-HCV infection were analyzed using DESeq2 (v1.38.3) [67]. Biological replicates were first assessed for quality by calculating Pearson's correlation coefficients (PCC) of read counts across genes. Any replicate displaying a PCC < 0.9 compared to the other replicates was removed before DEG analysis. Low-abundance genes were filtered out by retaining only those with at least 10 reads in at least 2 samples.

Differential expression analyses were conducted using raw read counts, as recommended by DESeq2, to compare the transduced versus mock-transduced samples at each time point. A gene was considered significantly differentially expressed if it had an adjusted p-value < 0.05 and an absolute log2 fold change > 2. Heatmaps were generated with the pheatmap (v1.0.12) package [68], using the Z-score of each gene for visualization. The top 25 upregulated genes identified in the mmRTP4-transduced versus mock-transduced comparison at 12 hours post-infection were selected and visualized in the heatmap.

## Western blot

Cells were centrifuged at 2000 rpm for 3 min at 4 °C, and the resulting pellets were lysed for 5 min on ice in RIPA buffer (50 mM Tris, pH 7–8; 0.1% (w/v) SDS; 0.5% (w/v) sodium deoxycholate; 1% (v/v) Triton-X-100) containing 1X protease inhibitor cocktail (Sigma Aldrich, St. Louis, MO, USA). Lysates were subsequently spun down at 15000 g for 60 min at 4 °C. Supernatants were mixed with 6X Laemmli buffer (375 mM Tris pH 7–8, 10% (w/v) SDS, 50% (v/v) glycerol, 10% (v/v) β-mercaptoethanol, 0.03% (w/v) bromophenol blue) and heated for 5 min at 95 °C. Protein concentration was quantified via Pierce BCA assay (Thermo Fisher Scientific, Waltham, MA, USA).

60 µg of each sample were separated on a 15% (w/v) SDS-polyacrylamide gel in running buffer (25 mM Tris, 192 mM glycine, 0.1% (w/v) SDS in deionized $H_2O$) at 150 V for 60 min. Proteins were then transferred onto a 0.2 µm nitrocellulose membrane (Bio-Rad Laboratories, Hercules, CA, USA) in transfer buffer (25 mM Tris, 192 mM glycine, 0.04% (w/v) SDS, 20% (v/v) methanol in deionized $H_2O$) at 95 V for 70 min. Membranes were treated with blocking solution (5% (w/v) milk in PBS) overnight at 4 °C. Membranes were washed twice with 0.05% (v/v) Tween-20 in PBS (PBS-T) for 5 min and incubated with primary antibodies diluted in PBS-T for 60 min at room temperature. Membranes were washed thrice with PBS-T for 5 min and incubated with secondary antibodies diluted in PBS-T for 30 min at room temperature, followed by three more washes with PBS-T. Membranes were subsequently visualized on an Odyssey CLx Imaging System (LI-COR Biotechnology, Lincoln, NE, USA).

## Proximity-ligation assay

HCV infections were conducted in a 6-well plate with 1.5E5 Huh7-Lunet cells seeded per plate pre-infection. Cells were infected with HCV (J6/JFH1-Jc1) at an MOI of 1. On day 1 post-infection, 3E4 cells were seeded onto collagen-coated coverslips in 24-well plates. The next day, cells were transduced with 500 µL 1:5 diluted lentivirus (FLAG-mmRTP4 or FLAG-hsRTP4) without spinoculation. At 48 hours post-transduction, cells were fixed and permeabilized in ice-cold 100% methanol for 30 min at −20 °C. The Duolink *In Situ* Detection Reagents Green (Sigma-Aldrich, St. Louis, MO, USA) kit was used following the manufacturer's instructions, with the following modifications. Replacement buffers were used for wash buffer A (150 mM NaCl, 10 mM Tris base, 0.05% Tween-20 in $H_2O$, pH 7.4), and wash buffer B (100 mM NaCl, 40 mM Tris base, 160 mM Tris-HCl in $H_2O$, pH 7.5) [69]. For secondary antibodies, Duolink *In Situ* PLA Probes Anti-Rabbit MINUS and Anti-Mouse PLUS were used. Coverslips were mounted in Duolink *In Situ* Mounting Medium with DAPI and imaged using a Nikon Ti-E microscope (Nikon, Melville, NY, USA) within the Princeton University Confocal Microscopy Facility. Images

were taken at 40X magnification and analyzed using Fiji (ImageJ2) image analysis software and Python. Foci were quantified using the Python OpenCV library; a binary threshold was applied to the image using the cv2.THRESH_OTSU method [70], contours were detected using cv2.findContours(), and objects with a total area below 10 pixels$^2$ were filtered out as noise. Foci were quantified for cells within 10 images per sample.

### *In vivo* experiments

#### Generation of CD81$^{EL2[H/H]}$ OCLN$^{EL2[H/H]}$ RTP4$^{-/-}$ mice

The generation of CD81$^{EL2[H/H]}$ OCLN$^{EL2[H/H]}$ has been described previously [31]. RTP4$^{-/-}$ mice were generated previously by targeting the first and second coding exons and were kindly provided by Xin-zhuan Su (NIAID) [30].

To generate CD81$^{EL2[H/H]}$ OCLN$^{EL2[H/H]}$ RTP4$^{-/-}$ mice, RTP4$^{-/-}$ mice were bred with CD81$^{EL2[H/H]}$ OCLN$^{EL2[H/H]}$ mice, and the progeny generation (CD81$^{EL2[M/H]}$ OCLN$^{EL2[M/H]}$ RTP4$^{+/-}$) was intercrossed. To test for CD81$^{EL2[H/H]}$ and OCLN$^{EL2[H/H]}$, we conducted qPCR on ear clippings as described previously [31]. To test for RTP4$^{-/-}$, we conducted 3 diagnostic PCR reactions to screen for the novel junction present in knockout mice. In wild-type mice, PCRs 1 and 3 produce a band, whereas PCR 2 produces a band in knockout mice. To extract DNA from ear punch biopsies, ear clips were boiled in 80 μL digestive buffer (6.25 mL 1M NaOH, 50 μL 0.5M EDTA, 118.7 mL H$_2$O, pH 12) for 1 h at 95 °C, and subsequently quenched with 80 μL neutralization buffer (40 mM Tris-HCl, pH 5). The resultant pH-neutral suspension of genomic DNA was utilized for genotyping PCRs. To extract RNA, we utilized the Monarch Total RNA Miniprep Kit (New England Biolabs, Ipswich, MA, USA) (Table 2).

### Mouse hepatocyte isolation and transplantation

Murine hepatocyte isolation and transplantation into FNRG mice were conducted as described previously [71]. In brief, mice were anaesthetized by intraperitoneal injection of a mixture of 100 mg/kg ketamine and 10 mg/kg xylazine. Livers were perfused through the portal vein with a chelating solution (0.01M HEPES pH 7.3 and 0.5 mM EGTA pH 8.0 in Ca$^{2+}$/Mg$^{2+}$-free EBSS) at a flow rate of 2mL/min until the liver bleached, followed by 40 ml collagenase solution (0.01M HEPES pH 7.3 and 1mg/ml Collagenase Type II in EBSS with Ca$^{2+}$, Mg$^{2+}$ and Phenol Red). The digested liver was cut into pieces,

**Table 2. Primers utilized for mouse genotyping in this study.**

| Primer | Purpose | Sequence (5'–3') |
| --- | --- | --- |
| PU-O-3812 | mCD81$^{hEL2}$ qPCR primer 1 | CCAAGGCTGTGGTGAAGACTTTC |
| PU-O-3814 | mCD81$^{hEL2}$ qPCR primer 2 | GGCTGTTCCTCAGTATGGTGGTAG |
| PU-O-3812 | mCD81$^{WT}$ qPCR primer 1 | CCAAGGCTGTGGTGAAGACTTTC |
| PU-O-3813 | mCD81$^{WT}$ qPCR primer 1 | TGTTCTTGAGCACTGAGGTGGTC |
| PU-O-1235 | mOCLN$^{hEL2}$ qPCR primer 1 | GTGTTTATTGCCACGATCGTGT |
| PU-O-1236 | mOCLN$^{hEL2}$ qPCR primer 2 | AAATTGGTTGCAGAGGGCATAT |
| PU-O-1237 | mOCLN$^{WT}$ qPCR primer 1 | CTCTTTGGAGGAAGCCTAAACTACC |
| PU-O-1238 | mOCLN$^{WT}$ qPCR primer 1 | AAACTGGTTGCAGATCATATAT |
| PU-O-1000 | mGAPDH qPCR primer 1 | ACGGCCGCATCTTCTTGTGCA |
| PU-O-1001 | mGAPDH qPCR primer 2 | ACGGCCAAATCCGTTCACACC |
| PU-O-9208 | RTP4$^{-/-}$ genotyping PCR 1 FP | TAGGTGATTAGGAACACAACC |
| PU-O-9209 | RTP4$^{-/-}$ genotyping PCR 1 RP | AGCGACCCTAACCATCTTAGC |
| PU-O-9208 | RTP4$^{-/-}$ genotyping PCR 2 FP | TAGGTGATTAGGAACACAACC |
| PU-O-9210 | RTP4$^{-/-}$ genotyping PCR 2 RP | GCTATTTTCAGAGCATGTCC |
| PU-O-9693 | RTP4$^{-/-}$ genotyping PCR 3 FP | GCAGAAGTTGGACCTCTGC |
| PU-O-9210 | RTP4$^{-/-}$ genotyping PCR 3 RP | GCTATTTTCAGAGCATGTCC |

transferred into a washing solution (0.01M HEPES pH7.3 and 10% FBS in DMEM), passed through a 100 μm cell strainer, washed and passed through a 100 μm cell strainer. The resulting cell suspension was passed through a 70 μm cell strained. Cell suspension was washed for three more times with spinning steps at 140g for 5 min to remove unwanted cellular contaminates. Cells were resuspended in HyClone DMEM (Cytiva, Marlborough, MA, USA) and cell viability was assessed through trypan blue exclusion.

## Engraftment of mouse hepatocytes into FNRG recipients

The generation of Fah$^{-/-}$ NOD.Cg-Rag1$^{tm1Mom}$IL2rg$^{tmlWjl}$/SzJ IL2Rg$^{null}$ (FNRG) mice has been previously described [32]. FNRG mice maintained on water supplemented with 10% (w/v) 2-(2-nitro-4-trifluoromethylbenzoyl)-1,3-cyclohexanedione (NTBC, Yecuris Inc., Tualatin, OR, USA), to block the build-up of metabolites to toxic concentrations. To facilitate hepatic engraftment female FNRG mice older than 6 weeks of age were injected intrasplenically with 1E6 hepatocytes freshly isolated from CD81$^{EL2[H/H]}$ OCLN$^{EL2[H/H]}$ RTP4$^{-/-}$ mice. Following transplantation FNRG mice were given water lacking NTBC for 9 days, followed by 7 days with 1% NTBC, 7 days with water lacking NTBC, and then 4 days with 1% NTBC. Following this, mice were solely provided water lacking NTBC.

## HCV RNA isolation from serum

Mouse blood was harvested by cheek-puncture at the aforementioned intervals. Serum was harvested by centrifuging the coagulated blood (3500 rpm, 10 min, room temperature) and collecting the supernatant. Total RNA was isolated from 25 μL serum using the Zymo Viral RNA extraction kit (Genesee Scientific, El Cajon, CA, USA) or the KingFisher Flex System (Thermo Fisher Scientific, Waltham, MA, USA), and the HCV genome copy number was quantified by one-step RT-qPCR using a Multicode-RTx HCV RNA kit (Luminex Corporation, Austin, TX, USA) and a StepOne Real Time PCR (Applied Biosystems, Waltham, MA, USA), according to manufacturer's instructions.

## HCV RNA isolation from liver tissue

Mouse livers were harvested postmortem in accordance with protocols reviewed and approved by the Institutional Animal Care and Use Committees (IACUC) of Princeton University. Liver tissue was stored at −80 °C in RNALater. Stainless steel beads (5 mm, Qiagen, Hilden, Germany) and 350 μL lysis buffer were added to sample tubes containing 10–50 mg liver tissue and homogenized using a TissueLyser LT (Qiagen, Hilden, Germany). Total RNA was isolated from lysate using the Monarch Total RNA Miniprep Kit (New England Biolabs, Ipswich, MA, USA) or the KingFisher Flex System (Thermo Fisher Scientific, Waltham, MA, USA), and the HCV genome copy number was quantified by one-step RT-qPCR using a Multicode-RTx HCV RNA kit (Luminex Corporation, Austin, TX, USA) and a StepOne Real Time PCR (Applied Biosystems, Waltham, MA, USA), according to manufacturer's instructions.

## Histology processing, chromogenic immunohistochemistry, and whole slide scanning

Tissue samples were fixed for a minimum of 72 h in 4% (w/v) paraformaldehyde (PFA) before processing in a Tissue-Tek VIP-5 automated vacuum infiltration processor (Sakura Finetek USA, Torrance, CA, USA) and embedded in paraffin using a HistoCore Arcadia paraffin embedding machine (Leica, Wetzlar, Germany). 5-μm tissue sections were generated using a RM2255 rotary microtome (Leica, Wetzlar, Germany) and transferred to positively charged slides. A Ventana Discovery Ultra tissue autostainer (Roche Diagnostics, Indianapolis, IN, USA) was used for chromogenic immunohistochemistry (IHC). A rabbit primary polyclonal antibody specific to fumarylacetoacetase (FAH) was diluted to 1/100 in Ventana antibody diluent with casein (Roche) and incubated with tissue samples at RT for 3 hours (Invitrogen: PA5–42049), followed by incubation with a secondary goat anti-rabbit HRP-polymer antibody (Vector Laboratories, Burlingame, CA, USA) for 20 min at 37ºC, and developed with 3,3-diaminobenzidine (DAB) and hematoxylin counterstain (Roche). Histology images were acquired using a PhenoImager whole slide scanner (Akoya Biosciences, Marlborough, MA, USA) for figure preparation.

**Table 3. Primers utilized for RT-qPCR quantification of ectopic RTP4 and housekeeping genes.**

| Primer | Purpose | Sequence (5'–3') |
|---|---|---|
| PU-O-8026 | hsHPRT1 qPCR primer 1 | ACTGAAGAGCTATTGTAATGACCAG |
| PU-O-8027 | hsHPRT1 qPCR primer 2 | TGGATTATACTGCCTGACCAAG |
| PU-O-10332 | FLAG-hs/mmRTP4 qPCR primer 1 | GATTACAAGGATGACGACGATAAG |
| PU-O-11974 | FLAG-hsRTP4 qPCR primer 2 | CTAGCTGAAGGTTGCCATCCAA |
| PU-O-12546 | FLAG-mmRTP4 qPCR primer 2 | ACAATGTTCTTATCCAAATGCAGGC |
| PU-O-10768 | mmRTP4 qPCR primer 1 | TGGGAGCAGACATTTCAAGAAC |
| PU-O-10769 | mmRTP4 qPCR primer 2 | ACCTGAGCAGAGGTCCAACTT |
| PU-O-11728 | mmHPRT1 qPCR primer 1 | TCAGTCAACGGGGGACATAAA |
| PU-O-11729 | mmHPRT1 qPCR primer 2 | GGGGCTGTACTGCTTAACCAG |

### Transfection of FLAG-RTP4 constructs

293T cells were seeded in 10 cm dishes at a density of 2.2E6 cells/plate. Huh7 Lunet cells were seeded in 6-well plates at a density of 3E5 cells/well. 12 hours post-seeding, cells were transfected with either 10 µg (293T) or 1.5 µg (Huh7 Lunet) of tagged constructs utilizing X-tremeGENE HP DNA Transfection Reagent (Roche Applied Science; Indianapolis, IN). Media was changed 24 hours post-transfection. 48 hours post-transfection, cells were harvested for downstream analyses.

### IFNβ stimulation of primary murine hepatocytes (PMHs)

PMHs were harvested from 6-8 week old C57BL/6 mice via collagenase perfusion as described above. PMHs were seeded into a collagen-coated 24-well plate at a density of 8E4 cells/well. 3 hours post-seeding, PMHs were treated with 250 IU recombinant mouse IFN-β1 (BioLegend, San Diego, CA, USA). 12 hours post-stimulation, total RNA was isolated for RT-qPCR analysis (see below).

### Quantification of ectopic RTP4 expression via RT-qPCR

RNA was harvested from cell pellets utilizing the Monarch Total RNA Miniprep Kit (New England Biolabs, Ipswich, MA, USA), following manufacturer instructions. Transcripts from the resultant eluates were quantified using Luna Universal qPCR Master Mix (New England Biolabs, Ipswich, MA, USA) with the primers listed in Table 3.

### Supporting information

**S1 Fig. Structural prediction of RTP4 chimeras. A-B.** AlphaFold structural predictions of **A.** hsRTP4 and **B.** mmRTP4. **C.** MatchMaker superimposition of structures in **A** and **B. D-E.** AlphaFold structural predictions of **D.** hsRTP4$^{mmDVR}$ and **E.** mmRTP4$^{hsDVR}$. **F.** MatchMaker superimposition of structures in **D** and **E. G-L.** AlphaFold structural predictions of **G.** hsRTP4$^{mmZFD\_1}$, **H.** hsRTP4$^{mmZFD\_2}$, **I.** hsRTP4$^{mmZFD\_3}$, **J.** mmRTP4$^{hsZFD\_1}$, **K.** mmRTP4$^{hsZFD\_2}$, **L.** mmRTP4$^{hsZFD\_3}$.
(DOCX)

**S2 Fig. Quantification of FLAG-RTP4 expression via flow cytometry, RT-qPCR, and Western blot. A.** Transfection of FLAG-tagged RTP4 bicistronic expression constructs in 293T cells. **B.** Quantification of bicistronic fluorophore expression in transfected samples. **C.** RT-qPCR analysis of samples using primers targeting FLAG and gene-specific sequences. Blue, FLAG-hsRTP4 RNA; green, FLAG-mmRTP4 RNA; ND, not detected. **D.** Western blot analysis of cell lysates using mouse anti-FLAG and rabbit anti-β-actin; this gel is utilized in Fig **5B**. Some figure elements (mouse [75], liver [76]) were sourced from the public domain and are listed as references.
(DOCX)

**S3 Fig. hsRTP4 ectopic expression *in vitro* is comparable to induced mmRTP4 expression in *ex vivo* tissue. A.** Schematic of FLAG-hsRTP4 transfection in Huh7 Lunet cells. **B.** RT-qPCR quantification of FLAG-hsRTP4 transgene expression in samples prepared in **A**. ND, not detected. **C.** Schematic of mmRTP4 induction in primary murine hepatocytes (PMH) *in vivo*. **D.** RT-qPCR quantification of FLAG-hsRTP4 transgene expression in samples prepared in **D**. (DOCX)

## Acknowledgments

The RTP4$^{-/-}$ mice were kindly provided by Xin-zhuan Su (NIAID), Huh7.5 cells, the anti-HCV NS5A mAb (clone 9E10) and the J6/JFH1 (Jc1) genome by Charles Rice (The Rockefeller University), Huh7.5.1 cells by Frank Chisari (The Scripps Research Institute), and Huh7-Lunet cells by Ralf Bartenschlager (University of Heidelberg). We also thank Wei Wang and other members of the Princeton University Genomics Core Facilities at the Lewis-Sigler Institute for Integrative Genomics for their help in assessing the RNA quality and maintaining and loading the Illumina NovaSeq. We thank members of the Ploss lab for critical discussions and comments on this project. We are grateful to Tina DeCoste from the Molecular Biology Flow Cytometry Resource Facility for excellent technical support. We would like to thank the Confocal Imaging Facility, a Nikon Center of Excellence, in the Department of Molecular Biology at Princeton University for instrument use and technical advice.

## Author contributions

**Conceptualization:** Michael P Schwoerer, Alexander Ploss.

**Data curation:** Michael P Schwoerer, Sebastian Carver, Aaron E Lin, Thomas R Cafiero, Keith A Berggren, Saori Suzuki, Brigitte Heller, Celeste Rodriguez, Aoife K O'Connell, Hans P Gertje, Nicholas A Crossland, Alexander Ploss.

**Formal analysis:** Michael P Schwoerer, Aaron E Lin, Jianche Liu, Serene Dhawan, Nicholas A Crossland, Alexander Ploss.

**Funding acquisition:** Nicholas A Crossland, Alexander Ploss.

**Investigation:** Michael P Schwoerer, Sebastian Carver, Aaron E Lin, Thomas R Cafiero, Keith A Berggren, Saori Suzuki, Brigitte Heller, Celeste Rodriguez, Aoife K O'Connell, Hans P Gertje, Nicholas A Crossland, Alexander Ploss.

**Methodology:** Michael P Schwoerer, Aaron E Lin, Nicholas A Crossland, Alexander Ploss.

**Project administration:** Nicholas A Crossland, Alexander Ploss.

**Resources:** Nicholas A Crossland, Alexander Ploss.

**Supervision:** Nicholas A Crossland, Alexander Ploss.

**Visualization:** Michael P Schwoerer, Jianche Liu, Serene Dhawan, Aoife K O'Connell, Hans P Gertje, Nicholas A Crossland, Alexander Ploss.

**Writing – original draft:** Michael P Schwoerer, Alexander Ploss.

**Writing – review & editing:** Michael P Schwoerer, Sebastian Carver, Aaron E Lin, Jianche Liu, Thomas R Cafiero, Keith A Berggren, Serene Dhawan, Saori Suzuki, Brigitte Heller, Celeste Rodriguez, Aoife K O'Connell, Hans P Gertje, Nicholas A Crossland, Alexander Ploss.

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
