## [Decision Letter · Decision Letter 0]

11 May 2025

Receptor transporter protein 4 (RTP4)-mediated repression of hepatitis C virus replication in mouse cells

PLOS Pathogens

Dear Dr. Ploss,

Thank you for submitting your manuscript to PLOS Pathogens. After careful consideration, we feel that it has merit but does not fully meet PLOS Pathogens's publication criteria as it currently stands. Therefore, we invite you to submit a revised version of the manuscript that addresses the points raised during the review process.

Please submit your revised manuscript within 60 days Jul 10 2025 11:59PM. If you will need more time than this to complete your revisions, please reply to this message or contact the journal office at plospathogens@plos.org. Please include the following items when submitting your revised manuscript:

We look forward to receiving your revised manuscript.

Kind regards,

Jin Zhong, Ph.D

Guest Editor

PLOS Pathogens

Michael Letko

Section Editor

PLOS Pathogens

Editor-in-Chief

PLOS Pathogens

orcid.org/0000-0003-2946-9497

Editor-in-Chief

PLOS Pathogens

orcid.org/0000-0002-7699-2064

**Additional Editor Comments :**

Your manuscript was fully evaluated at the editorial level and by three independent peer reviewers. The reviewers appreciated the attention to an important problem in the HCV research field, but raised some substantial concerns about the manuscript as it currently stands. We therefore ask you to modify the manuscript according to the review recommendations before we can consider your manuscript for acceptance. Your revisions should address the specific points made by each reviewer. Reviewers 1 and 2 asked for several additional experiments. In particular, I would like to ask you to address their comments on the interaction of HCV NS5A and RTP4, as well as the endogenous mmRTP4 expression in murine hepatocytes.

**Journal Requirements:**

- ® on pages: 9, 10, 12, and 13.14

- TM on pages: 9, and 13.

Potential Copyright Issues:

i) Figures 1A, 3A, 4A, 6A, and 7A. Please confirm whether you drew the images / clip-art within the figure panels by hand. If you did not draw the images, please provide (a) a link to the source of the images or icons and their license / terms of use; or (b) written permission from the copyright holder to publish the images or icons under our CC BY 4.0 license. Alternatively, you may replace the images with open source alternatives. See these open source resources you may use to replace images / clip-art:

5) When completing the data availability statement of the submission form, you indicated that you will make your data available on acceptance. We strongly recommend all authors decide on a data sharing plan before acceptance, as the process can be lengthy and hold up publication timelines. Please note that, though access restrictions are acceptable now, your entire data will need to be made freely accessible if your manuscript is accepted for publication. This policy applies to all data except where public deposition would breach compliance with the protocol approved by your research ethics board. If you are unable to adhere to our open data policy, please kindly revise your statement to explain your reasoning and we will seek the editor's input on an exemption. Please be assured that, once you have provided your new statement, the assessment of your exemption will not hold up the peer review process.

2) If any authors received a salary from any of your funders, please state which authors and which funders.

7) Please ensure that the funders and grant numbers match between the Financial Disclosure field and the Funding Information tab in your submission form. Note that the funders must be provided in the same order in both places as well.

**Reviewers' Comments:**

Reviewer's Responses to Questions

**Part I - Summary**

Reviewer #1: In this study, Ploss A et al. investigate the species-specific antiviral activity of receptor transporter protein 4 (RTP4), focusing on the black flying fox orthologue, with particular emphasis on the murine (mmRTP4) versus human (hRTP4) variants in the context of hepatitis C virus (HCV) infection. The authors demonstrate that ectopic expression of mmRTP4, but not hRTP4, significantly inhibits HCV infection, identifying the zinc-finger domain (ZFD) of mmRTP4 as essential for this antiviral activity. Key observations include enhanced interaction between mmRTP4 and HCV NS5A, as well as reduced NS5A protein levels in HCV-infected cells compared to hRTP4. Additionally, the authors attempted to disrupt RTP4 expression in mice with humanized alleles of CD81 and occludin (OCLN) but did not detect significant viral replication. While this work represents a significant effort toward establishing a murine model for HCV infection by modulating RTP4, the underlying mechanisms of HCV restriction in mice remain incompletely understood. Further exploration of the mechanistic basis of mmRTP4’s antiviral activity would strengthen the manuscript, particularly given the well-documented antiviral role of RTP4.

Reviewer #2: Schwoerer et al. explore the role of murine RTP4 in restricting HCV replication in mice. A previous study identified the ISG RTP4 as a species specific restriction factor for flaviviruses, including inhibitory effects of murine (mm)RTP4 on HCV. Building on this observation and using ectopic expression of hs- vs. mmRTP4 in Huh7-cells as a main model, the authors confirm the strong inhibition of HCV replication by mmRTP4. The authors generate chimeric RTP4 variants to identify the region underlying the phenotype and show that mmRTP4 twice as often found in proximity to NS5A than hsRTP4. mmRTP4 transiently upregulates a series of genes in Huh7 cells, but does not induce interferon, the functional significance of this data is so far not clear. Based on the hypothesis that RTP4 might be a major restriction factor for HCV replication in mice, the authors use RTP4 knockout mice and transfer RTP4 deficient murine hepatocytes in mice lacking adaptive immunity. However, unfortunately, RTP4 knockout does not increase permissiveness of mice to HCV infection.

The lack of small animal models is still a major roadblock in HCV research, particularly towards vaccine development. While this study cannot improve the efficiency of existing murine models, it adds additional data suggesting that the reasons for non-permissiveness of mice is based on multiple factors that need to be overcome. The data is overall convincing and of high quality and provides substantial novelty. However, some claims are not sufficiently supported by the data provided, this includes the interpretation of data on the chimeric RTP4 proteins, on claiming specific interaction with NS5A and on specific roles of RTP4 in virion formation.

Reviewer #3: In this manuscript the authors explore the role of receptor transporter protein 4 as restriction factor for HCV replication in mouse cells. Through substitution experiments in Huh7 cells with the murine and human alleles the study accounts for the restriction on HCV replication con-ferred by the murine allele of RTP4 (mmRTP4) in Huh7 cells. Detailed domain substitution ex-periments between human and murine alleles identify the restricting domain of RTP4 to reside within the zinc finger domain of the murine molecule. The authors further interrogate individual nucleic acid binding arrays within the zinc finger domain by substitution analysis and establish that tertiary structure is responsible for antiviral activity of the murine RTP4.

The authors further document that murine mmRTP4 abrogates established infections when transduced into infected cells. The authors probe, based on infection suppression in trans, whether this observation is relying on interferon signaling. However, by careful expression analysis of cells transduced with either mmRTP4 or the human ortholog hsRTP4 no differences in interferon signaling between the two variants can be documented. The authors conclude based on these observations that interferon signaling is not responsible for the replication sup-pressing function of mmRTP4.

Based on these observations, the authors next probe if direct interaction between RTP4 and HCV NS5A could be documented. Indeed, mmRTP4 displays a greater degree of association with NS5A as assessed by proximity ligation than does hsRTP4. These observations suggest mmRTP4 could have a direct effect on replication of HCV.

To interrogate this suggestion, HCV infection was assessed in HCV permissive mice human-ized for essential CD81 and occludin host factors while rendered deficient for RTP4.

However, HCV replication following infection of this mouse strain or a control humanized CD81 and occludin RTP4 sufficient strain did not document that absence of RTP4 enhanced replication. Further to eliminate the possibility that adaptive immune response in the mouse could affect the outcome, the authors transplanted RTP4 deficient hepatocytes from mice with humanized CD81 and occludin into a FAH−/− NOD Rag1−/− Il2rgNULL (FNRG) host. Here in the absence of adaptive immunity, RTP4 deficiency is not sufficient to support productive rep-lication of HCV. Based on this thorough evaluation the authors conclude that while restricting for HCV replication, mmRTP4 is not the sole murine host factor obstructing HCV replication in the mouse.

Significance:

This study and the experimental work on which it is based is of very high quality. The authors provide solid evidence that mouse mmRTP4 has a restricting effect on HCV replication and document that this is conveyed through interaction between NS5A and the mmRTP4 which displays a higher degree of interaction than the human ortholog hsRTP4. The authors further explore the suggestion that the restriction of replication by RTP4 could be alleviated by ablat-ing the molecule in vivo. However, as convincingly demonstrated in two experiments RTP4 deficiency is not solely able to support replication of HCV in mice.

Whereas the results of this study do not resolve mouse restricting HCV factors, the importance of the observations are nevertheless essential to the field of HCV research, which can pave the way for further exploration of replication restriction of HCV in the mouse.

**Part II – Major Issues: Key Experiments Required for Acceptance**

Reviewer #1: 1. Figure 2: The observation that mmRTP4hsDVR exhibited reduced antiviral activity compared to mmRTP4 suggests that the DVR-TM region is necessary but not sufficient for antiviral activity, as hsRTP4mmDVR lacked antiviral effects. Conversely, mmRTP4hsZFD_1 retained full activity, indicating that the C-terminal ZFD is both necessary and sufficient for antiviral function. Thus, based on these data, to this reviewer, the C-terminal part of ZFD and the DVR-TM are important for antiviral activity. Utilizing AlphaFold-predicted structures could provide further insights into these functional domains.

2. Figures 3C and 3D: The experimental details require clarification. Specifically, it is unclear whether cells were passaged during the 9-day period following HCVcc infection and lentiviral transduction. Given the sensitivity of HCV replication to cellular confluency, the observed phenotypes may be influenced by cell state. Additionally, the reduction in intracellular virus titer (Figure 3E) was less pronounced than the decrease in NS5A+ cells (Figure 3C), suggesting that mmRTP4 may primarily affect NS5A protein levels through translational or post-translational mechanisms. The reviewer disagrees with the conclusion that mmRTP4 inhibits infectious particle formation, as viral titers in the supernatant were comparable across groups.

3. Figure 5: The rationale for selecting a highly sensitive method to detect RTP4-NS5A interactions should be explained. The use of complementary techniques, such as co-immunoprecipitation (co-IP) or co-localization studies, would strengthen the findings. Additionally, demonstrating RTP4’s interaction with HCV replication complexes (RCs) through dsRNA detection using anti-dsRNA antibodies could provide further mechanistic insights. Or the MOA is the same as reported, that RTP4 binds to viral genome?

4. Figure 6B: It is unclear whether viral replication was detectable in the CD81EL2[H/H] OCLNEL2[H/H] control group. This information is critical for interpreting the results.

5. Figure 7: The absence of a positive control, such as IFN receptor knockout (IFNR KO) mice, limits the interpretability of the data. Including such controls would enhance the robustness of the findings.

6. Restriction Factor Phenotypes: The expected outcome of knocking out a restriction factor is enhanced viral replication. The reviewer suggests exploring the effects of hRTP4 knockout on human viruses and mmRTP4 knockout on murine viruses (e.g., murine hepatitis virus, MHV) to better understand the role of RTP4 in viral restriction.

Reviewer #2: 1. RTP4 is an ISG, but the entire manuscript uses ectopic expression of hs- or mmRTP4 in human cells as a model. Still, it would be important to know, how the levels of ectopic expression, particularly of mmRTP4, compare to native expression levels in murine hepatocytes, in presence and absence of interferon induction. If no antibodies are available, RNA-data could be used as an alternative.

2. The authors should provide bulk Western blot data to ensure similar expression of the RTF4 orthologues and chimeras, if antibodies are available, since the stability of the proteins might differ, particularly regarding the chimeras. This could at least be shown on protein level for the tagged variants used in Fig. 5.

3. Fig. 5 and associated text: The data indicate that RTP4 is in proximity to NS5A but neither that NS5A physically interacts nor that NS5A is the sole interacting protein in the HCV replicase, as claimed by the authors throughout the manuscript, including abstract. To support these claims, the authors need to include co-immunoprecipitation data (direct interaction) and use another protein of the replicase (e.g. NS3, NS4B or NS5B) as a control. Alternatively, the authors need to tone down their claims, e.g. by stating that NS5A was used as a marker for the replicase.

Reviewer #3: There are no major issues with the execution of this study. No further experiments required.

**Part III – Minor Issues: Editorial and Data Presentation Modifications**

Reviewer #1: (No Response)

Reviewer #2: 1. Fig. 1C and 2B: The way this data is normalized to non-transduced cells is not fully clear to this reviewer. Does the data contain any information/correlation to the relative expression of RTP4 or the respective autofluorescent protein as a correlate?

2. Fig. 2, lines 140-160: I do not fully agree with the interpretation of data by the authors. Replacement of the murine DRV/TM domain by the hs counterpart dramatically impacts on the restricting function of mmRTP4 (construct 2 compared to M). Therefore, the conclusion that the ZFD is the only determinant of the difference is in my view not supported by the data. The only chimera widely resembling the restricting function of mmRTP4 is construct 6, harboring the N-terminus of the hhZFD, while constructs 7 and 8 have no restricting capacity. Overall, the data at this point support auxiliary roles of mmDRV/TM.

3. Fig. 4: Panels B and C are not referred to in the text. The data should be interpreted more explicitely, regarding the nature of induced genes. The text mentions non-infected controls, how do this data look in comparison? Are the genes induced also in absence of HCV infection? Is any of the induced gene products secreted and could account for the trans-inhibitory effect observed in Fig. 3D?

4. Fig. 5D: IF data on NS5A should be included to show that NS5A is expressed to similar levels in both conditions.

5. Fig. 6D/E, 7D: How can HCV RNA be quantified up to several logs below LLOD? Formally, everything below LLOD should be non-detectable.

6. The authors claim throughout the manuscript, that mmRTP4 impacts on virion formation. However, mmRTP4 has no impact on extracellular titers and reduces intracellular titers (Fig. 3E/F). This phenotype could be fully explained by reduced RNA replication. Such mechanistic claims need to be supported by additional data (e.g. measuring HCV intracellular RNA, which should be unaffected, or using replicons) or avoided, e.g. by using more general terms (…impacts on viral replication)

Reviewer #3: The legend to Figure 3 is not easy to decipher and could with benefit to the reader be reworked for better clarity. Moreover, the data point in green (panel C) is only legible at very high magni-fication due to similar values as other data points. Means to mitigate this obstruction could also improve the figure.

PLOS authors have the option to publish the peer review history of their article (what does this mean? ). If published, this will include your full peer review and any attached files.

**Do you want your identity to be public for this peer review?** For information about this choice, including consent withdrawal, please see our Privacy Policy .

Reviewer #1: No

Reviewer #2: No

Reviewer #3: No

**Figure resubmission:**

**Reproducibility:**



---

## [Decision Letter · Decision Letter 1]

27 Jul 2025

Dear Dr. Ploss,

We are pleased to inform you that your manuscript 'Receptor transporter protein 4 (RTP4)-mediated repression of hepatitis C virus replication in mouse cells' has been provisionally accepted for publication in PLOS Pathogens.

Best regards,

Jin Zhong, Ph.D

Guest Editor

PLOS Pathogens

Michael Letko

Section Editor

PLOS Pathogens

Sumita Bhaduri-McIntosh

Editor-in-Chief

PLOS Pathogens

orcid.org/0000-0003-2946-9497

Michael Malim

Editor-in-Chief

PLOS Pathogens

orcid.org/0000-0002-7699-2064

Reviewer Comments (if any, and for reference):

Reviewer's Responses to Questions

**Part I - Summary**

Reviewer #1: All my concerns have been addressed

Reviewer #2: The authors have addressed all my specific points.

Reviewer #3: (No Response)

**Part II – Major Issues: Key Experiments Required for Acceptance**

Reviewer #1: (No Response)

Reviewer #2: (No Response)

Reviewer #3: (No Response)

**Part III – Minor Issues: Editorial and Data Presentation Modifications**

Reviewer #1: (No Response)

Reviewer #2: (No Response)

Reviewer #3: (No Response)

PLOS authors have the option to publish the peer review history of their article (what does this mean? ). If published, this will include your full peer review and any attached files.

**Do you want your identity to be public for this peer review?** For information about this choice, including consent withdrawal, please see our Privacy Policy .

Reviewer #1: No

Reviewer #2: No

Reviewer #3: No

---

## [Editor Report · Acceptance letter]

Dear Dr. Ploss,

We are delighted to inform you that your manuscript, "Receptor transporter protein 4 (RTP4)-mediated repression of hepatitis C virus replication in mouse cells," has been formally accepted for publication in PLOS Pathogens.

Best regards,

Sumita Bhaduri-McIntosh

Editor-in-Chief

PLOS Pathogens

orcid.org/0000-0003-2946-9497

Michael Malim

Editor-in-Chief

PLOS Pathogens

orcid.org/0000-0002-7699-2064